# Mucosal immunization with the lung *Lactobacillus*-derived amphiphilic exopolysaccharide adjuvanted recombinant vaccine improved protection against *P. aeruginosa* infection

**Haochi Zhang**[1,2☯], **Shouxin Sheng**[1☯], **Chunhe Li**[1], **Xuemei Bao**[1], **Lixia Zhao**[2], **Jian Chen**[1,2], **Pingyuan Guan**[2], **Xiaoyan Li**[2], **Na Pan**[1], **Yanchen Liang**[1], **Xueqi Wang**[1], **Jingmin Sun**[1], **Xiao Wang**[1] *

1 State Key Laboratory of Reproductive Regulation & Breeding of Grassland Livestock, School of Life Sciences, Inner Mongolia University, Hohhot, P.R. China, 2 The Spirit Jinyu Biological Pharmaceutical Co. Ltd, Hohhot, Inner Mongolia, China

☯ These authors contributed equally to this work.

* wangxiao@imu.edu.cn

## Abstract

Respiratory infections caused by *Pseudomonas aeruginosa* are a major health problem globally. Current treatment for *P. aeruginosa* infections relies solely on antibiotics, but the rise of antibiotic-resistant strains necessitates an urgent need for a protective vaccine. Traditional parenteral vaccines, despite employing potent adjuvants aimed at serotype-dependent immunity, often fail to elicit the desired mucosal immune response. Thus, developing vaccines that target both localized mucosal and systemic immune responses represents a promising direction for future research on *P. aeruginosa* vaccination. In this study, we explored EPS301, the exopolysaccharide derived from the lung microbiota strain *Lactobacillus plantarum* WXD301, which exhibits excellent self-assembly properties, enabling the formation of homogeneous nanoparticles when encapsulating recombinant PcrV of *P. aeruginosa*, designated as EPS301@rPcrV. Notably, the EPS301 vector effectively enhanced antigen adhesion to the nasal and pulmonary mucosal tissues and prolonged antigen retention. Moreover, EPS301@rPcrV provided effective and sustained protection against *P. aeruginosa* pneumonia, surpassing the durability achieved with the "gold standard" cholera toxin adjuvant. The EPS301-adjuvanted vaccine formulation elicited robust mucosal IgA and Th17/γδ17 T cell responses, which exceeded those induced by the CTB-adjuvanted vaccination and were sustained for over 112 days. Additionally, Th 17 and γδ 17 resident memory T cells induced by EPS301@rPcrV were crucial for protection against *P. aeruginosa* challenge. Intriguingly, IL-17A knockout mice exhibited lower survival rates, impaired bacterial clearance ability, and exacerbated lung tissue damage upon EPS301 adjuvanted vaccination against *P. aeruginosa*-induced pneumonia, indicating an IL-17A-dependent protective mechanism. In conclusion, our findings provided direct evidence that

**Data Availability Statement:** All relevant data are provided as figures within the paper and its Supporting information files.

**Funding:** X. W. was supported by the Science and Technology Major Project of the Inner Mongolia Autonomous Region of China (grant no. 2021ZD0013), the Key Scientific and Technological Research Program of Inner Mongolia Autonomous Region (grant no. 2021GG0156), the National Natural Science Foundation of China (grant no. 32060800) and the Technology Major Project of Hohhot (2023-JieBangGuaShuai). H.C.Z. was supported by the China Postdoctoral Science Foundation (grant no. 2023MD734180). The funders had no role in study design, data collection and analysis, decision to publish, or preparation of the manuscript.

**Competing interests:** The authors have declared that no competing interests exist.

EPS301@rPcrV mucosal vaccine is a promising candidate for future clinical application against *P. aeruginosa*-induced pulmonary infection.

## Author summary

Antibiotics currently dominate *P. aeruginosa* treatment, but rising resistance demands a protective vaccine. Yet, traditional injectable vaccines, even with potent adjuvants targeting serotype immunity, often fall short in inducing the crucial mucosal immune response. Mucosal vaccines excel at inducing antigen-specific protective immune responses at the mucosal barrier, while simultaneously stimulating systemic humoral and cell-mediated immune responses throughout the body. Leveraging its exceptional self-assembly properties, EPS301 has emerged as a promising nasal mucosal adjuvant for vaccines, providing effective and sustained protection against *P. aeruginosa*-induced pneumonia. While EPS301-adjuvanted vaccination elicited robust lung IgA and serum IgG responses, humoral immunity alone did not confer significant protection. Notably, the protective effect significantly diminished in IL-17A knockout mice, but not IFN-γ knockout mice, suggesting a crucial role for IL-17A in EPS301-mediated immunity. Additionally, intranasal co-administration of EPS301 with rPcrV antigen induced strong lung Th17/γδ17 resident memory T cells, which effectively protected against *P. aeruginosa*. Our findings highlight EPS301 as a promising mucosal adjuvant for developing intranasal vaccines against *P. aeruginosa* pneumonia.

## Introduction

Acute lung injury and acute respiratory distress syndrome (ALI/ARDS) pose a critical global health threat, claiming countless lives annually. Pneumonia, particularly from viral and bacterial infections, remains a leading cause of infectious death. The ongoing COVID-19 pandemic, driven by the highly contagious SARS-CoV-2 virus, primarily manifests as ALI/ARDS and has impacted nearly every nation worldwide. This viral infection cripples the immune system, leaving individuals vulnerable to secondary bacterial infections, further complicating the already critical illness [1–2]. The secondary bacterial infection always renders poor prognosis of the patients. *P. aeruginosa* is a significant cause of healthcare-associated infections, being particularly problematic in intensive care units [3]. Its infections are associated with high morbidity and mortality in many groups, including individuals with healthcare-associated pneumonia, chronic obstructive pulmonary disease (COPD), or cystic fibrosis (CF) [4–5]. It is included in the "critical" category of the World Health Organisation's (WHO) priority list of bacterial pathogens for which research and development of new therapy methods are urgently needed. *P. aeruginosa*-induced infections are difficult to treat because of their complex pathogenic mechanisms, and the ability of this bacterium to quickly become resistant to various anti-biotics [2,6]. Therefore, an effective *P. aeruginosa* vaccine could have an important role both prophylactically in preventing acquisition of infection and, possibly in combination with other treatments in a therapeutic context to prevent re-infections. Numerous attempts have been made during the past five decades to develop an effective *P. aeruginosa* vaccine, but success to date is quite limited [7]. Recent study demonstrated that PcrV is highly conserved and well-validated to be a good target for immunoprophylactic strategies against *P. aeruginosa* in animal models [8–9]. Indeed, the PcrV protein form a ring structure located at the tip of the

type 3 secretion system (T3SS) and play a crucial role in facilitating effector translocation as well as bacterial pathogenicity [10].

While most licensed vaccines rely on injections, mucosal vaccines excel at inducing protective mucosal immune responses that block infection or transmission at the entry point [11]. Designing effective mucosal vaccines, or even determining their necessity, requires upfront consideration of the infection's nature. For enteric pathogens, for example, invasiveness varies: some like typhoid and polio invade deeper tissues, while shigella partially invades, and cholera and ETEC remain strictly mucosal [12–13]. This dictates the desired immune response: solely mucosal, primarily mucosal with some systemic support, or purely systemic. Additionally, the specific mucosal surface (e.g., inflamed vs. uninflamed) influences antibody accessibility, dominant isotype, and transport mechanisms for reaching pathogens [14–16]. Mucosal vaccines can effectively induce protective immune responses against pathogens at the mucosal site, and also induce antigen-specific humoral and cell-mediated immune responses throughout the body [17]. Although mucosal vaccines are generally more effective than injectable vaccines in stimulating local protection in the respiratory and gastrointestinal tracts, there are currently few mucosal vaccines available for human and animal use. This is largely due to a lack of efficient, non-toxic adjuvants that enhance immune response for mucosal vaccine delivery [18]. Firstly, the adjuvant of the mucosal vaccine should have the function of increasing immune efficacy and protecting antigens from proteolytic degradation [19]. Secondly, mucosal vaccine adjuvants required functions of enhancing T lymphocyte response and generations of multi-faceted amplified immune responses without compromising safety [20]. It remains challenging for achieving these goals with recent mucosal adjuvants.

The utilization of biodegradable polymers as an adjuvant to package antigens into nano-vehicles for mucosal vaccine preparation is a prospective strategy to improve immune responses sustainably release antigens and protect the loaded antigen from degradation [21–23]. Natural biopolymers, such as pullulan, alginate, inulin, chitosan, and microbial exopolysaccharides, provide an optimized strategy to break the limitation of synthesized polymers [24–26]. The nature biopolymers have properties of low-toxicities and biodegradable. For example, chitosan-based nanoparticles have particularly been widely studied for their biocompatibility, biodegradability, nontoxicity, and easy modification into ideal shapes and sizes [27]. However, natural biopolymer delivery systems require multiple modification processes to provide better water solubility and stability. Alameh and Jiang reported, the increase of hydrophobic substituents increased the particle size, encapsulation rate and drug loading of chitosan, while hydrophilic substituents provide better water solubility and stability of nanoparticles systems [28]. In addition, Solmaz reported, applying alginate polymer as a nasal delivery carrier was not considered immunogenic against influenza whole virus [29]. Overall, the superior properties of naturally occurring nano-scale biomaterials facilitate their diverse applications in delivery systems.

Microbes and their byproducts are increasingly recognized for their ability to regulate host health and immune function. Among these, exopolysaccharides (EPS), high-molecular-weight sugar polymers secreted by microorganisms, have garnered significant interest. We have been examining the biological functions of EPS produced by *Lactobacillus* and found that these EPS as an adjuvant induced IFN-γ production by splenocytes [30]. In particular, *Lactobacillus plantarum* WXD301, which isolated from lung commensal bacteria, derived exopolysaccharide (EPS301) nanocarriers possess the ability to deliver antigens and elicit efficient antigen-specific antitumor immunity in solid tumor models (data have not published). In this study, our findings provided direct evidence that a novel mucosal nano-vaccine, EPS301@rPcrV, enhancing protective lung Th17 and γδ 17 T responses, is a promising candidate for future clinical application against *P. aeruginosa*-induced pulmonary infection.

## Results

### Self-assembly characteristics of EPS301

Pulmonary infection caused by *P. aeruginosa* has created an urgent need for an efficient vaccine, but the protection induced by current candidates is limited, partially lack of proven vaccine-effective antigens and safe mucosal adjuvants to enhance local immune responses as well as the lack of accepted correlates of protection [11]. Natural polysaccharides are the best biopolymers candidate for vaccine adjuvants because which reported to have self-assemble properties and immunity-enhancing abilities [31]. The secretory exopolysaccharides, named EPS301 was isolated from commensal bacteria *L. plantarum* WXD301 by ethanol precipitation methods. Further purified by a combination of ion-exchange chromatography and gel permeation chromatography as described previously [32]. As shown in S2A and S2B Fig, lyophilized EPS301 showed a flocculent structure and EPS301 aqueous solution was colorless with good dispersion. According to the HPGPC analysis, EPS301 was observed as single and symmetrically sharp peak in the HPLC chromatogram (S2C Fig). This characteristic suggested its homogeneity based on the Mw distribution. The molecular weight of HSP-W was calculated as 84.3 kDa. EPS301 consisted of glucosamine hydrochloride, galactosamine hydrochloride, galactose, glucose, mannose and glucuronic acid with a molar ratio of 4.3:4.1:1.4:1:4.8:7.6 (S2D Fig). The scanning electron microscopy (SEM) images revealed that EPS301 possessed a fibrous structure (S2E Fig). After homogenization, the EPS301 self-assembled into spherical nanoparticles (Fig 1A). Dynamic light scattering (DLS) studies showed the mean size of the particles was around 24 ± 6 nm (polymer dispersion index (PDI):0.142 ± 0.06) at room temperature, bright and relatively homogenous in water (Fig 1A). According to the self-assemble properties of EPS301, the EPS301 was mixed with *P. aeruginosa* antigen (rPcrV) to prepare nanoparticles after homogenization. The SEM and TEM images showed that EPS301 based nanoparticles was well-dispersed with uniform spherical morphology (Figs S2E and 1A). The size of EPS301@rPcrV nanoparticles was 43.10 ± 5 nm (polymer dispersion index (PDI):0.121 ± 0.04) (Fig 1B). The stability of EPS301 in PBS was confirmed for 21 days (S2F Fig). Besides, the stability of EPS301@rPcrV nanoparticles in various solvents, including PBS, serum, and cell medium, was confirmed for 21 days (S2G Fig). Additionally, using Zeta potentials test, we found that EPS301 (-38.47 ± 0.53), rPcrV (-20.83 ± 0.19) and EPS301@rPcrV (-17.05 ± 0.24) were negatively charged (Fig 1C).

The mucosal membrane, composed of a mucus gel layer and a single layer of epithelial cells, acts as a natural barrier that protects the host by rapidly capturing and clearing foreign particles. To effectively elicit mucosal immune responses and avoid mucus clearance, a mucosal vaccine needs to adhere to the mucosa and prolong antigens retention [33–34]. We labeled rPcrV with Cy7 and compared the retention times of the antigen alone versus those formulated with EPS301 following intranasal administration (Fig 1D). As shown in Fig 1E, in the group of mice immunized with rPcrV alone, rPcrV rapidly disappeared from nasal cavity within 12 h. In sharp contrast, the EPS301 based nanoparticle prolonged the retention of rPcrV in the nasal cavity, as the signal was still detectable at 96 h. A much stronger signal of rPcrV was observed in lung of EPS301@rPcrV inoculated mice than rPcrV inoculated alone. The fluorescence signal in major tissues, nasal cavity, lung, heart, liver, spleen and kidney was detected at 9 h, 48 h and 96 h post vaccination. As shown in Fig 1F, the fluorescent signal could still be detected in the lungs 96 hours after intranasal immunization with EPS301@PcrV. However, in the mice immunized with rPcrV alone, the lung fluorescence signal had almost disappeared 48 hours after immunization, indicating that EPS301, as a mucosal adjuvant, can significantly enhance antigen adhesion to mucosal tissues and prolong antigen release. We observed fluorescence signals in the liver of the mice after immunization, which may be due to

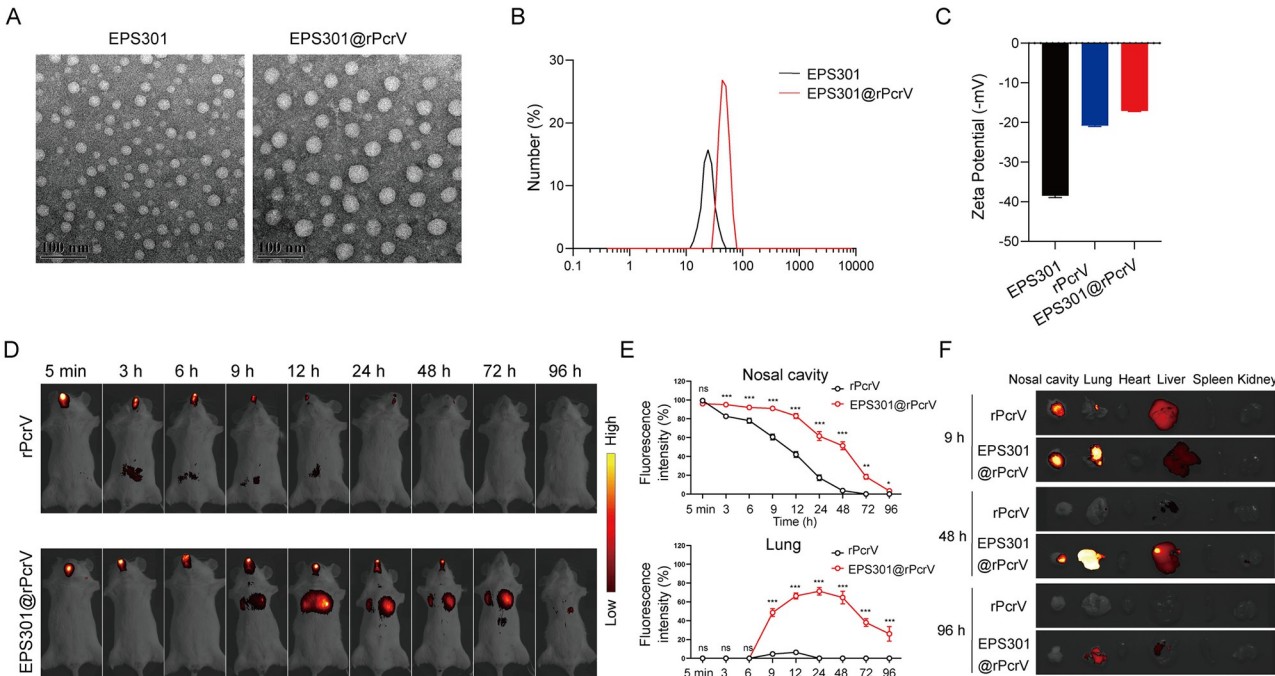

**Fig 1. Characteristics of EPS301@rPcrV nanoparticles.** (A) Representative TEM images of EPS301 and EPS301@rPcrV particles. Scale bars, 100 nm. (B) Size distributions of EPS301 and EPS301@rPcrV. DLS results showed the size of self-assembled EPS301 about 24 ± 6 nm, EPS301@rPcrV about 43.10 ± 5 nm. (C) The Zeta potentials of EPS301, rPcrV and EPS301@rPcrV. Zeta potentials results showed EPS301 about -38.47 ± 0.53, rPcrV about -20.83 ± 0.19, EPS301@rPcrV about -17.05 ± 0.24. (D) Representative in vivo fluorescence images of mice at the indicated time points after intranasal administration of free rPcrV or EPS301@ rPcrV. rPcrV in both groups labeled with Cy7. (E) Relative fluorescence intensity of rPcrV or EPS301@rPcrV in mouse nasal cavity and lung. (F) Representative ex vivo fluorescence images of major tissues (nasal cavity, lung, heart, liver, spleen and kidney) at 9 h, 48 h and 96 h after intranasal administration of free rPcrV or EPS301@ rPcrV. rPcrV in both groups labeled with Cy7. Significant differences were calculated with One- or Two-way ANOVA followed by Tukey's multiple comparisons test. ns, not significant, *p < 0.05, **p < 0.01, ***p < 0.001. Data are presented as means ± SEM.

liver metabolism. To further demonstrate the general applicability of EPS301 in prolonging antigen release, we labeled OVA with Cy7 and compared the retention times of the antigen alone versus those formulated with EPS301 following intranasal administration. As shown in the S3H Fig, we found the same results that nanoparticles based on EPS301 significantly prolonged the antigen release. These results suggested that the EPS301 exhibits excellent self-assembly properties, forming a homogeneous nano-vaccine when encapsulated with antigens. Moreover, intranasal vaccination with EPS301 encapsulation could effectively enhance antigen adhesion to the nasal and pulmonary mucosal tissues, thereby prolonging antigen retention.

### Intranasal vaccination with a EPS301-adjuvanted subunit vaccine induced effective protection against *P. aeruginosa* pneumonia

For exploring the potential of EPS301 as the effective mucosal adjuvant to *P. aeruginosa* infection, mice were immunized twice by airway (intranasal, i.n.) administration routes with a recombinant *P. aeruginosa* antigen, rPcrV and EPS301 or, the "gold standard" mucosal adjuvant, the cholera toxin B subunit (CTB) (Fig 2A). 7 days after the booster immunization, we sacrificed the mice and collected their lungs for H&E staining to assess the safety of EPS301. As shown in S3 Fig, there was no significant inflammation in the lungs of the mice in all immunized group before infection. Mice was inoculated with bacterial slurry (1×10⁹ CFUs of

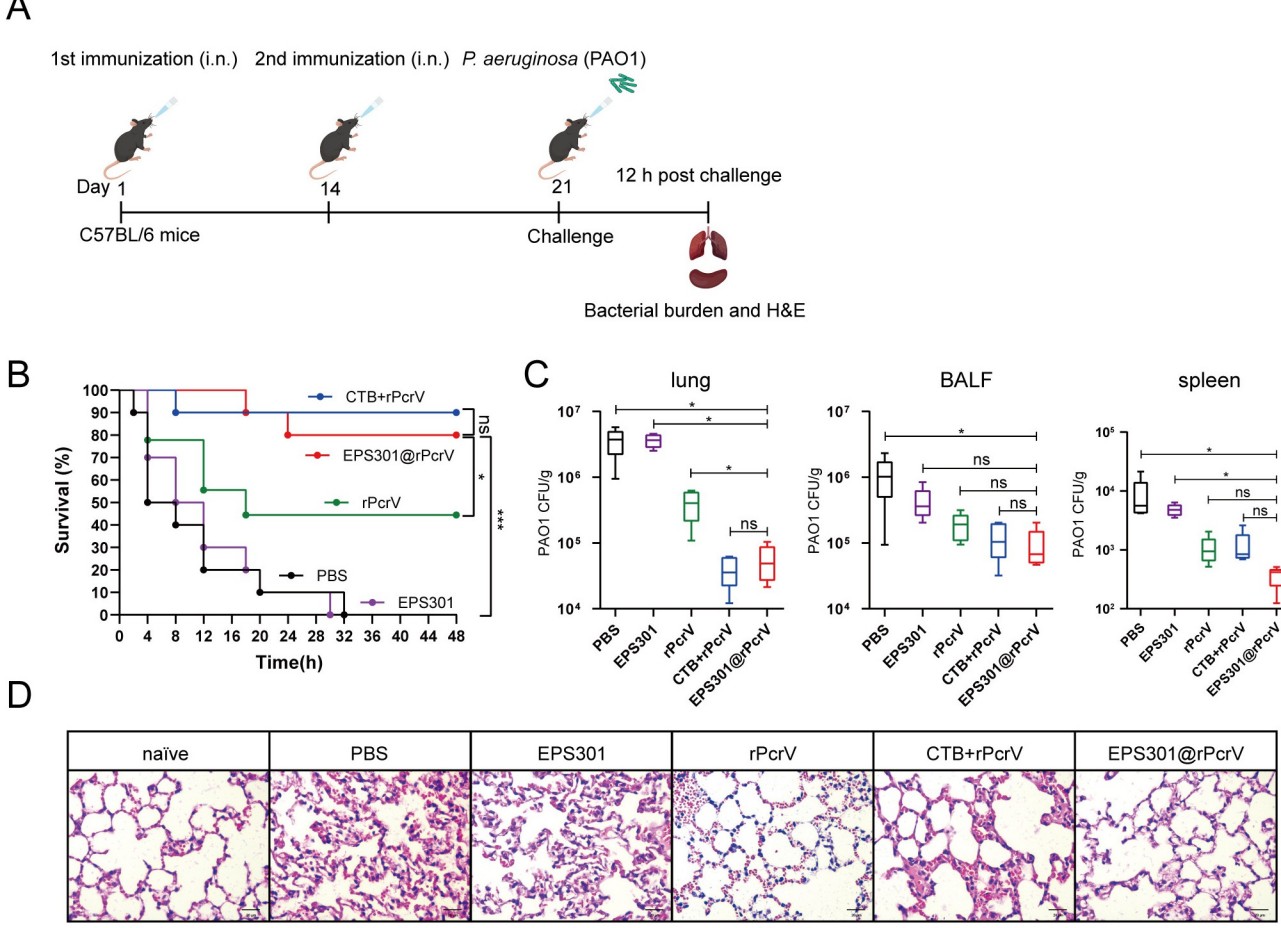

**Fig 2. EPS301-adjuvanted vaccination induced effective protection against *P. aeruginosa* pneumonia.** (A) Experimental design and timeline of prime and booster regimen of PBS, EPS301, rPcrV, CTB+rPcrV and EPS301@rPcrV vaccination (i.n.) and challenge. (B) Mice were inoculated with 40 μl bacterial slurry ($1\times10^9$ CFUs of *P. aeruginosa* PAO1) via left nostril on day 7 post booster vaccination and held upright for 1 min. Representative survival rates from two independent experiments are shown (n = 8–10). The data for survival test were analyzed by Wilcoxon log-rank survival test (*p < 0.05; ***p < 0.001; ns, not significant). (C) Mice were inoculated with 40 μl bacterial slurry (lower dose, $1\times10^7$ CFUs of *P. aeruginosa* PAO1) as above. At 12 h post challenge, the numbers of bacteria in lungs, BALF and spleens were counted (n = 5–8). (D) Mice were inoculated with the lower dose of PAO1 as above. At 12 h post challenge, histological evaluation of lung sections by light microscopy. Lung specimens were fixed, sectioned, and stained with H&E (n = 3–5). Data are presented as means ± SEM. Significant differences were calculated with One-way ANOVA followed by Tukey's multiple comparisons test. ns, not significant, *p < 0.05, ***p < 0.001.

*P. aeruginosa* PAO1) via i.n. on day 7 post booster vaccination and monitored for up to 48 h. As shown in Fig 2B, EPS301-adjuvanted vaccination resulted in the higher protection against *P. aeruginosa* induced pneumonia (approximately 80% survival rates) compared with non-vaccinated mice (PBS and EPS301 alone inoculation) and rPcrV alone vaccinated mice, comparable to the survival rates of the mice vaccinated with CTB+rPcrV. EPS301@rPcrV induced resistance was further evaluated by treatment with a lower dose of *P. aeruginosa* PAO1 ($1\times10^7$ CFUs) that caused morbidity but not death, allowing assessment of bacterial loads and lung pathology. Intranasal immunization with EPS301@rPcrV led to a significant reduction of bacterial loads in lung, BALF and spleen compared with the non-vaccinated group, comparable to the levels of reduction seen in mice immunized with rPcrV adjuvanted with CTB (Fig 2C). Aggregates of dark purple-stained immune cells and lung architecture were preserved in

vaccinated mice after 12 h infection. In contrast, lungs of non-vaccinated mice showed notable damage (Fig 2D). Additionally, The CTB+rPcrV group of mice exhibited more severe lung damage and heightened inflammation compared to the EPS301 adjuvanted group following infection with *P. aeruginosa*. These data indicated that EPS301-adjuvanted intranasal vaccination regimen provided effective protection to mice against *P. aeruginosa* pneumonia.

## EPS301-adjuvanted intranasal vaccination provided sustained protection against *P. aeruginosa* pneumonia

To assess the long-term protective immune responses induced by EPS301, we first examined whether EPS301-adjuvanted vaccine could elicit strong and long-term rPcrV-specific IgA in lung homogenate and IgG in serum (Fig 3A). As shown in Fig 3B, intranasal immunization with rPcrV adjuvanted with EPS301 resulted in a significantly elevated rPcrV-specific IgA titer in lung already from day 14, which was maintained for at least 16 weeks post booster vaccination. Notably, antigen specific lung IgA responses induced by EPS301@rPcrV from day 42 after the second round of immunization were even higher than those generated by vaccination with the antigen and CTB. We also demonstrated that the combination of *P. aeruginosa* antigen and adjuvant was needed for antigen-specific IgA; thus, immunization with rPcrV alone gave almost no IgA in lung. Then, we detected the level of IgG in serum post vaccination. As shown in Fig 3C, EPS301@rPcrV immunization gave markedly increased secretion of IgG, 10–100-fold upward the responses observed with the mice immunized with rPcrV alone, comparable to the level of IgG in serum of the mice immunized with CTB+rPcrV. To further evaluate the duration of immunity provided by the EPS301@rPcrV, mice were challenged with *P. aeruginosa* 112 days after booster vaccination (Fig 3A). As shown in Fig 3D, intranasal immunization with rPcrV adjuvanted with EPS301 provided the survival reached 60%, significantly higher than non-vaccinated mice and rPcrV alone immunized mice, even higher than the mice immunized with the rPcrV+CTB. Additionally, bacterial loads in organs, particularly in lung were significantly reduced in the EPS301-adjuvanted vaccination group than other groups (Fig 3E). EPS301-adjuvanted vaccinated mice shown the intact pneumonocytes and clear cell structures post *P. aeruginosa* infection (Fig 3F). These findings, therefore, indicated that EPS301-adjuvanted intranasal vaccination could elicit strong and long-term mucosal IgA and systemic IgG, as well as promote sustained resistance to *P. aeruginosa*-induced pneumonia.

## EPS301-adjuvanted intranasal vaccination induced strong T-cell responses both in lung and spleen

T-cell responses and particularly Th 17 cells have previously been shown to play a crucial role in vaccine-induced protection against *P. aeruginosa* infection using curdlan-adjuvanted formulations [35]. To assess vaccine-induced T-cell responses, mice were immunized intranasally in two rounds with *P. aeruginosa* antigen alone or antigen adjuvanted with EPS301 or CT, and cellular responses in lung and spleen were analyzed. As shown in Fig 4A, the number and percentage of IFN-γ-secreting CD4 T cells in lung of EPS301-adjuvanted mice was significantly accumulated compared with non-vaccinated mice, but much below than CTB-adjuvanted mice on day 7 post vaccination in response to infection. Strikingly, the greater number and percentage of Th 17 cells was provoked in the mice vaccinated with EPS301, even surpassing that in CTB-adjuvanted mice after the infection on day 7 post vaccination (Fig 4A). Although CD4 T cells, and their production of IFN-γ and IL-17, have been extensively investigated in *P. aeruginosa* infection and are considered important vaccine targets, these cells are not the exclusive producers of these cytokines [36]. In fact, multiple lymphoid cell types have been

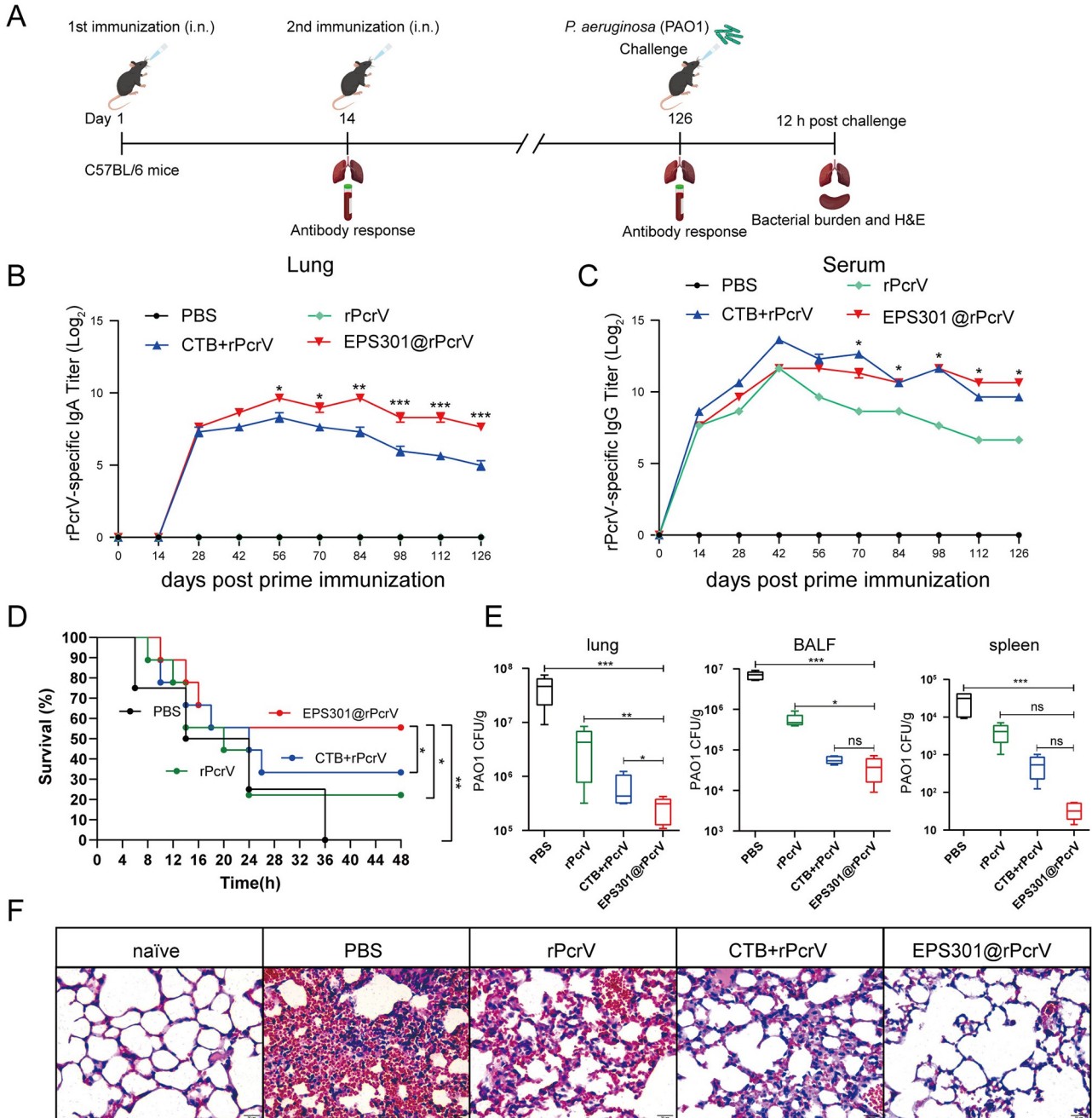

**Fig 3. Sustained protection and humoral immune responses in mice adjuvanted with EPS301.** (A) Experimental design and timeline of prime and booster regimen of PBS, rPcrV, CTB+rPcrV and EPS301@rPcrV vaccination (i.n.) and challenge. rPcrV-specific IgA titers in lung homogenate and IgG titers in serum were determined by ELISA before each round of immunization and every 14 days after the primary immunization. (B) The graph line shown the evolution of IgA titers in lung homogenate post immunization for each immunized group of mice. *p < 0.05; **p < 0.01; ***p < 0.001 for comparison with CTB+rPcrV immunized mice (n = 8). (C) The graph line shown the evolution of IgG titers in serum post immunization for each immunized group of mice. *p < 0.05 for comparison with rPcrV alone immunized mice (n = 8). (D) Mice were inoculated with 40 μl bacterial slurry (1×10$^9$ CFUs of *P. aeruginosa* PAO1) via left nostril on day 112 post booster vaccination and held upright for 1 min. Representative survival rates from two independent experiments are shown (n = 8–10). The data for survival test were analyzed by Wilcoxon log-rank survival test (*p < 0.05; **p < 0.01). (E) Mice were inoculated with 40 μl bacterial slurry (lower dose, 1×10$^7$ CFUs of *P. aeruginosa* PAO1) as above. At 12 h post challenge, the numbers of bacteria in lungs, BALF and spleens were counted (n = 5–8). (F) Mice were inoculated with the lower dose of PAO1 as above. At 12 h post challenge, histological evaluation of lung sections by light microscopy. Lung specimens were fixed, sectioned, and stained with H&E (n = 3–5). Data are presented as means ± SEM. Significant differences were calculated with One- or Two-way ANOVA followed by Tukey's multiple comparisons test. ns, not significant, *p < 0.05, **p < 0.01, ***p < 0.001.

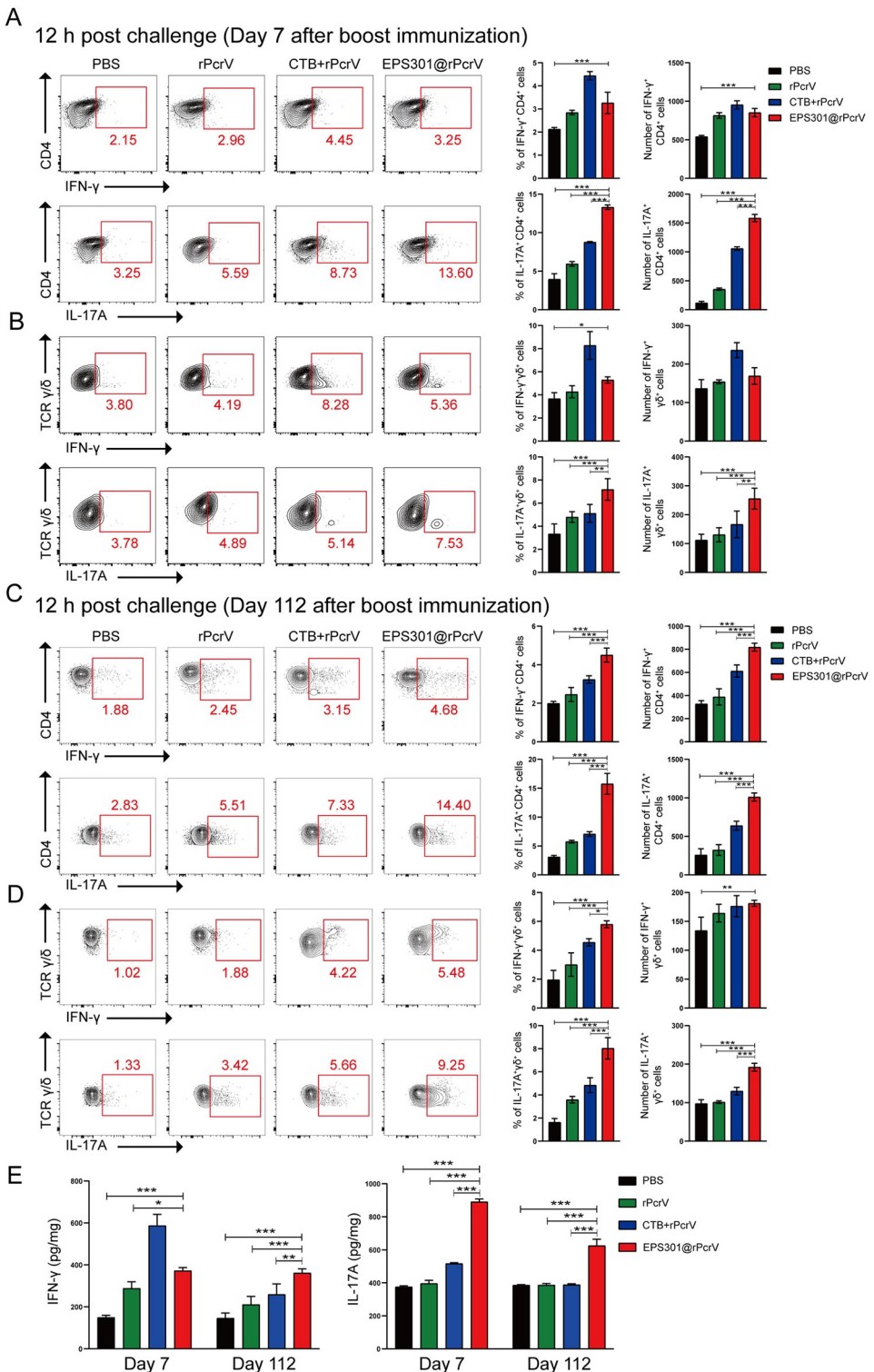

**Fig 4. Cellular immune responses in lung induced by intranasal immunization adjuvanted with EPS301.** Mice (n = 3-5/group) were immunized (i.n.) twice 14 days apart with rPcrV, CTB+ rPcrV or EPS301@rPcrV, with animals receiving PBS served as controls. Mice were inoculated with 40 μl bacterial slurry (lower dose, 1×10⁷ CFUs of *P. aeruginosa* PAO1) as above. Vaccinated mice were sacrificed at 12 hours post-challenge on day 7 or day 112 after the second vaccination lung tissue were prepared. Number of IFN-γ⁺ CD4⁺ T cells, IL-17A⁺ CD4⁺ T cells (A), IFN-γ⁺ γδ⁺ T cells, IL-17A⁺ γδ⁺ T cells (B) in lung at 12 hours post-challenge on day 7 after the second vaccination were estimated

by intracellular cytokine. Number of IFN-γ⁺ CD4⁺ T cells, IL-17A⁺ CD4⁺ T cells (C), IFN-γ⁺ γδ⁺ T cells, IL-17A⁺ γδ⁺ T cells (D) in lung at 12 hours post-challenge on day 112 after the second vaccination were estimated by intracellular cytokine. IFN-γ and IL-17 levels, determined by ELISA in a supernatant of lung tissue homogenate were analyzed. The IFN-γ levels and IL-17A levels (E) in lung at 12 hours post-challenge on day 7 and day 112 after the second vaccination were determined by ELISA. Data are presented as means ± SEM. Significant differences were calculated with One-way ANOVA followed by Tukey's multiple comparisons test. *p < 0.05; **p < 0.01; ***p < 0.001 for comparison with EPS301@rPcrV immunized mice.

discovered whose functions can overlap with those of conventional T cells, such as γδ T cells. We, therefore, measured IFN-γ-secreting and IL-17A-secreting γδ T cells after infection on day 7 post vaccination in response to infection. As shown in Fig 4B, the number and percentage of IFN-γ⁺ γδ⁺ T cells in lung of EPS301-adjuvanted mice was significantly higher than that in non-vaccinated mice on day 7 post vaccination, but below than CTB+rPcrV. An increased number of IL-17A⁺ γδ⁺ T cells were observed in the lung of the mice adjuvanted with EPS301. However, there were no significant differences in lung CD8 T cells among all groups of animals (S4 Fig). We next assessed the long-term immunity induced by EPS301. As shown in Fig 4C and 4D, the number and percentages of IFN-γ⁺ CD4⁺ T cells, IFN-γ⁺ γδ⁺ T cells, and particularly IL-17A⁺ CD4⁺ T cells and IL-17A⁺ γδ⁺ T cells in lung of EPS301-adjuvanted mice was markedly higher than that in other groups on day 112 post vaccination in response to infection. The cellular responses in spleen were consistent with the lung post immunization in response to infection (S5 Fig). Cytokine related to Th1-immune response (IFN-γ) was significantly elevated in EPS301@rPcrV immunized mice compared to rPcrV alone immunized mice and non-vaccinated mice in response to an infection on day 7 post booster vaccination both in lung and spleen (Figs 4E and S5E). On the other hand, it was observed that vaccination with the EPS301-adjuvanted rPcrV significantly enhanced IL-17A compared with other groups in response to an infection both on day 7 and day 112 post vaccination (Figs 4E and S5E). These results demonstrated that intranasal adjuvanted with EPS301 could provoke greater short-term and long-term cellular immune responses both in lung and in spleen.

## EPS301-adjuvanted induced mucosal CD4⁺ T cells, but not systemic IgG, are essential for protection against pulmonary *P. aeruginosa* infection

As described above, we found that EPS301-adjuvanted vaccination not only elicited significant long-term antigen-specific pulmonary IgA and systemic IgG responses but also markedly enhanced pulmonary and splenic cellular responses. However, it had reported that antibody responses were required to prevent bacterial dissemination but dispensable for lung-specific immunity [37], we next asked whether the enhanced *P. aeruginosa* antigen-specific systemic IgG response induced by the EPS301-adjuvanted vaccination was efficiency for *P. aeruginosa* pneumonia. The opsonophagocytosis assay assessed the correlation between functional antibody levels in serum samples and protection. The preincubation of only undiluted HI serum from rPcrV, CTB+rPcrV and EPS301@rPcrV vaccinated groups significantly enhanced bacterial uptake by RAW 264.7 murine macrophage cells as compared to the responses observed with PBS serum (Fig 5A). However, there were no significant differences in serum opsonophagocytosis after 10 or 100-fold dilution. The results suggested that antibodies from vaccinated mice exhibited opsonizing bactericidal activity against *P. aeruginosa* in a concentration-dependent manner. To further investigate whether serum from immunized mice can provide protection against *P. aeruginosa* infection, naïve C57BL/6 recipient mice were passively transferred with individual pooled serum from non-vaccinated (PBS), or rPcrV-, rPcrV+CTB- and EPS301@rPcrV-vaccinated mice. The antigen-specific IgG in serum of recipient mice was

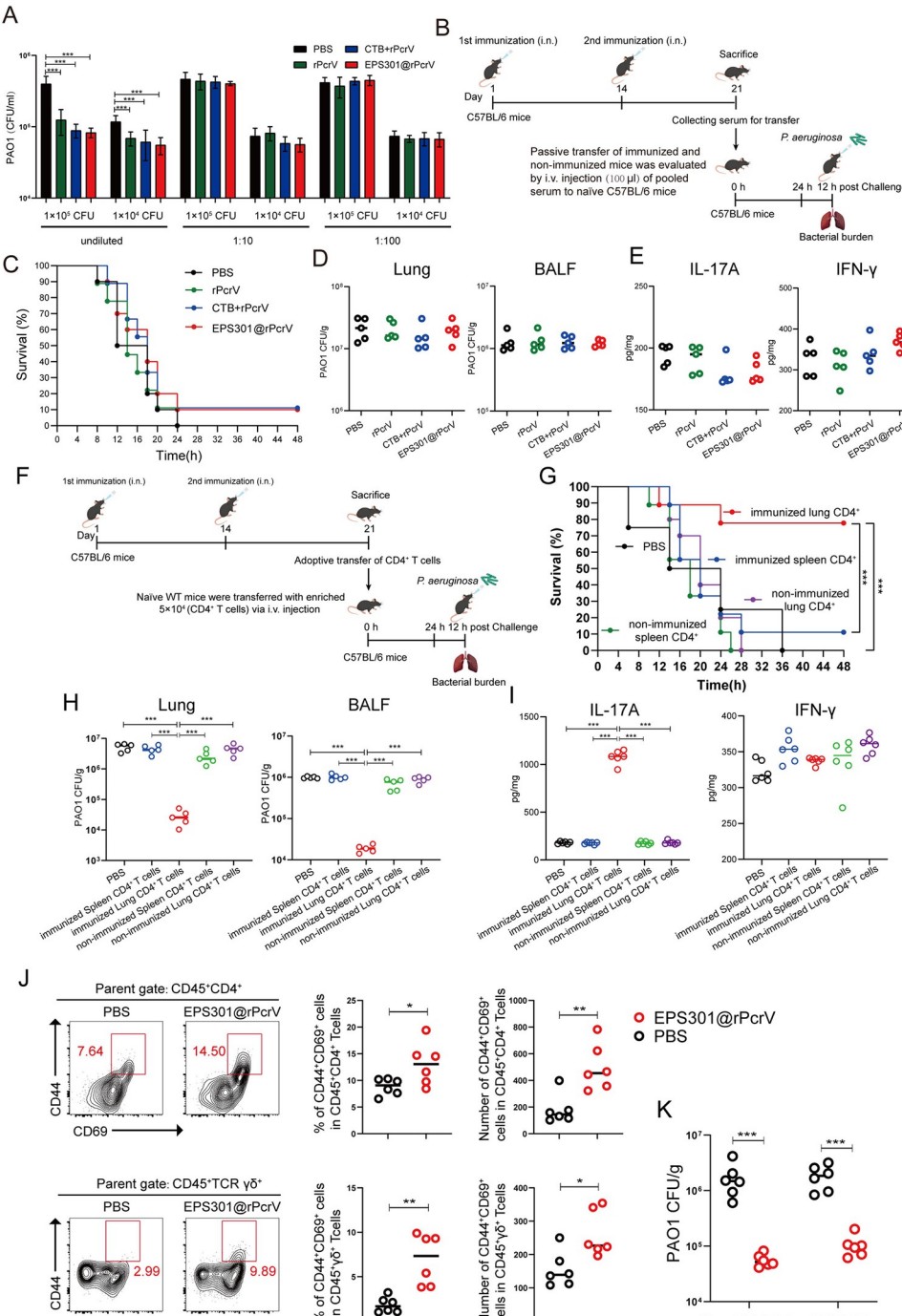

**Fig 5. EPS301-adjuvanted induced mucosal CD4⁺ T cells are essential for protection against pulmonary *P. aeruginosa* infection.** (A) Comparative analysis of opsonphagocytosis against PAO1 using antisera from differently immunized mice. (B) Timeline of passive transfer and *P. aeruginosa*-induced pneumonia model. (C) Passive transfer of immunized and non-immunized mice was evaluated by i.v. injection of pooled serum (100 μl) to naïve C57BL/6 mice (n = 10/group). 24 hours after serum transfer, mice were inoculated with 40 μl bacterial slurry ($1 \times 10^9$ CFUs of *P. aeruginosa* PAO1) via intranasal route (i.n.). Survival of mice from PBS, rPcrV, rPcrV+CTB and EPS301@rPcrV groups were observed for 48 h. The data for survival test were analyzed by Wilcoxon log-rank survival test (ns, not significant). (D) A lower dose of bacterial slurry ($1 \times 10^7$ CFUs of *P. aeruginosa* PAO1) was inoculated to recipient mice via intranasal route (i.n.). Bacterial loads in lung and BALF were detected at 12 h post infection. (E) 12 hours post-challenge, lung tissue was prepared. IL-17A and IFN-γ levels determined by ELISA in a supernatant of lung tissue homogenate were analyzed. (F) Timeline of adoptive transfer and *P. aeruginosa*-induced pneumonia model. Lung and

splenic CD3$^+$CD4$^+$ T cells from EPS301@rPcrV–vaccinated congenic mice were purified on day 7 after second vaccination. CD3$^+$CD4$^+$ T cells (5×10$^4$) were intravenously transferred into naïve C57BL/6 mice. (G) 24 hours after adoptive transfer, mice were inoculated with 40 µl bacterial slurry (1×10$^9$ CFUs of *P. aeruginosa* PAO1) via intranasal route (i.n.). Survival of mice were observed for 48 h. Survival after *P. aeruginosa* challenge (data were pooled from two independent experiments (n = 8–10). The data for survival test were analyzed by Wilcoxon log-rank survival test (***p < 0.001). (H) A lower dose of bacterial slurry (1×10$^7$ CFUs of *P. aeruginosa* PAO1) was inoculated to recipient mice via intranasal route (i.n.). Bacterial loads in lung and BALF were detected at 12 h post infection. (I) 12 hours post-challenge, lung tissue was prepared. IL-17A and IFN-γ levels determined by ELISA in a supernatant of lung tissue homogenate were analyzed. (J) Coexpression of CD44 and CD69 on non-stimulated lung CD4$^+$ or γδ$^+$ T cells (gated in CD45$^+$CD4$^+$ T cells). (K) FTY720 was initially dissolved in DMSO to create a 50 mg/mL stock solution. This stock solution was subsequently diluted with saline for intraperitoneal administration at a dosage of 30 µg per mice, and with distilled water for administration via stomach intubation at a dosage of 50 µg per mice. A lower dose of bacterial slurry (1×10$^7$ CFUs of *P. aeruginosa* PAO1) was inoculated to FTY720 treated and FTY720 non-treated mice via intranasal route (i.n.). Bacterial loads in lung were detected at 12 h post *P. aeruginosa* infection. Data are presented as means ± SEM. Significant differences were calculated with One-way ANOVA followed by Tukey's multiple comparisons test (D, E, H and L), Mann-Whitney U test (J) or unpaired t test (K). *p < 0.05, **p < 0.01, ***p < 0.001.

detected and compared among vaccination groups. Results showed significant increases of antigen-specific IgG in serum of recipient mice passive transferred from rPcrV, CTB+rPcrV and EPS301@rPcrV compared to the non-vaccinated (PBS) group, but much lower than those in actively immunized mice (S6 Fig). Additionally, a passive transfer experiment was performed where mice received pooled serum through an intravenous injection (i.v.) with subsequent challenge with 1×10$^9$ CFUs of *P. aeruginosa* PAO1 via the intranasal route (Fig 5B). As shown in Fig 5C, the serum from animal inoculated with PBS, rPcrV, rPcrV+CTB and EPS301@rPcrV provided noneffective protection against *P. aeruginosa* challenge, with below 10% survival. In addition, passive transfer of systemic IgG also insufficient in reducing bacterial loads both in lung and BALF (Fig 5D). Levels of IL-17A and IFN-γ after challenge were shown no significant difference among these groups (Fig 5E).

To determine the protective roles of T cells, we performed adoptive transfer of lungs and spleens CD3$^+$CD4$^+$ T cells from EPS301-vaccinated mice into naïve mice before *P. aeruginosa* PAO1 challenge as further validation (Fig 5F). As shown in Fig 5G and 5H, transferred immunized lung CD3$^+$CD4$^+$ T cells but not splenic CD3$^+$CD4$^+$ T cells efficiently enhanced the survival and reduced control of bacterial growth after *P. aeruginosa* challenge. Strikingly, a significant increase in IL-17A expression was measured in lung tissue from mice transferred to lung CD3$^+$CD4$^+$ T cells compared with other groups, whereas no significant increase in IFN-γ expression was observed post infection (Fig 5I). Several papers have shown that lung tissue-resident memory (TRM) cells generated from Th 17 cells can provide prompt protection to bacterial pneumonia [38]. Combining with these results, we conjectured that EPS301-adjuvanted vaccine may elicit lung TRM cells immune response. As shown in Fig 5J, day 7 post booster immunization, immunized lung CD4$^+$ T cells expressed CD44 and CD69, which are considered markers for lung TRM cells [38–39]. TRMs are not depleted by the administration of the drug FTY720 (fingolimod) [40]. Therefore, we treated EPS301@rPcrV vaccinated mice with FTY720 and then challenged with PAO1. As shown in Fig 5K, EPS301@rPcrV vaccination resulted lower bacterial loads post infection both in FTY720 treated and FTY720 untreated mice, which also proved that EPS301 adjuvanted vaccine induced lung TRMs are important for anti-infection. Taken together, these data indicated that despite the significant increase in *P. aeruginosa* antigen-specific antibody generated by vaccination, the humoral immune components in the blood of immunized animals did not provide enhanced protection compared to control as a standalone correlate. It is lung CD4$^+$ T cells induced by EPS301 formation vaccine that is essential for protection against pulmonary *P. aeruginosa* infection. Additionally, EPS301 can induce lung TRM immune response.

## EPS301-adjuvanted vaccination induced lung γδ17 T responses are critical for bacterial control

Traditional adjuvants, MF59, CFA/IFA, CTB induced adaptive cellular responses such as Th1 and Th17. However, in this study, we showed for the first time that a mucosal adjuvant formulation could not only provoke CD4 T cells mediate immune response, but also induced strong γδ T cells mediate immune response, which may be essential against *P. aeruginosa* pneumonia. γδ T cells are generally recognized as innate immune cells, but actually play key roles in both innate and adaptive immune function [41]. To further assess the function of EPS301-adjuvanted γδ T response, we first detected Th- 17 cells in WT and TCR δ-deficient mice before immunization, after immunization, and following infection. As shown in Fig 6A, there was no significant difference in the number and percentage of Th 17 cells between TCR δ-deficient and WT mice before immunization. However, the number and percentage of Th 17 cells were much higher in WT mice than TCR δ-deficient mice both after immunization and post infection. To some extent, this also proved that γδ T cells is a bridge between innate and adaptive immune response. Additionally, TCR δ-deficient mice were vaccinated with EPS301@rPcrV and survival, bacterial loads and the lung pathology were detected on day 7 post second vaccination. As shown in Fig 6B, EPS301-adjuvanted vaccine immunized TCR δ-deficient mice provided noneffective protection against *P. aeruginosa* challenge with much lower survival compared with vaccinated WT mice. EPS301@rPcrV immunized TCR δ-deficient mice also provided no significant efficiency reducing *P. aeruginosa* loads in lung and BALF (Fig 6C). Histological evaluation 12 h post-infection revealed complete pneumonocyte structure and substantial air space in vaccinated WT mice, whereas vaccinated TCR δ-deficient mice showed major alveolar damage and infiltration of large numbers of immune cells (Fig 6D). These results proved that EPS301-adjuvanted γδ T response is essential against *P. aeruginosa* pneumonia. To determine the protective roles of γδ T cells, we performed adoptive transfer of lungs and spleens CD3$^+$γδ T cells from EPS301@rPcrV-vaccinated mice into naïve mice before *P. aeruginosa* PAO1 challenge as further validation (Fig 6E). As shown in Fig 6F and 6G, transferred immunized lung CD3$^+$γδ T cells but not splenic CD3$^+$γδ T cells efficiently enhanced the survival and reduced control of bacterial growth after *P. aeruginosa* challenge. Strikingly, a significant increase in IL-17A expression was measured in lung tissue from mice transferred to lung CD3$^+$γδ T cells compared with other groups, whereas no significant increase in IFN-γ expression was observed post infection (Fig 6H). Meanwhile, immunized lung γδ$^+$ T cells expressed CD44 and CD69 (Fig 5J). These data demonstrated that mucosal γδ T cells mediate immune response induced by the EPS301-adjuvanted vaccination critical for anti-*P. aeruginosa* pneumonia.

## EPS301-adjuvanted vaccination induced protection against *P. aeruginosa* pneumonia in an IL-17A–dependent manner

To further assess the possibility that EPS301 -adjuvanted vaccination induced production of these cytokines promoted resistance to *P. aeruginosa*-induced pneumonia, IL-17A-deficient, IFN-γ-deficient and WT mice were vaccinated with EPS301@rPcrV (Fig 7A). 7 days post 2$^{nd}$ vaccination, serum and lung of the mice were collected for analyzing the levels of IgG and IgA. As shown in Fig 7B and 7C, strikingly, a significant increase of IgA titer was measured in the lung from vaccinated IFN-γ-deficient mice and vaccinated WT mice compared with vaccinated IL-17A-deficient mice, whereas no significant difference of IgG in serum was observed. Survival rate was ~90% for vaccinated WT mice, ~80% for vaccinated IFN-γ-deficient mice, much lower (~0%) for non-vaccinated control mice and (~10%) for vaccinated IL-17A-deficient mice (Fig 7D). Following *P. aeruginosa* infection, vaccinated IL-17A-deficient mice

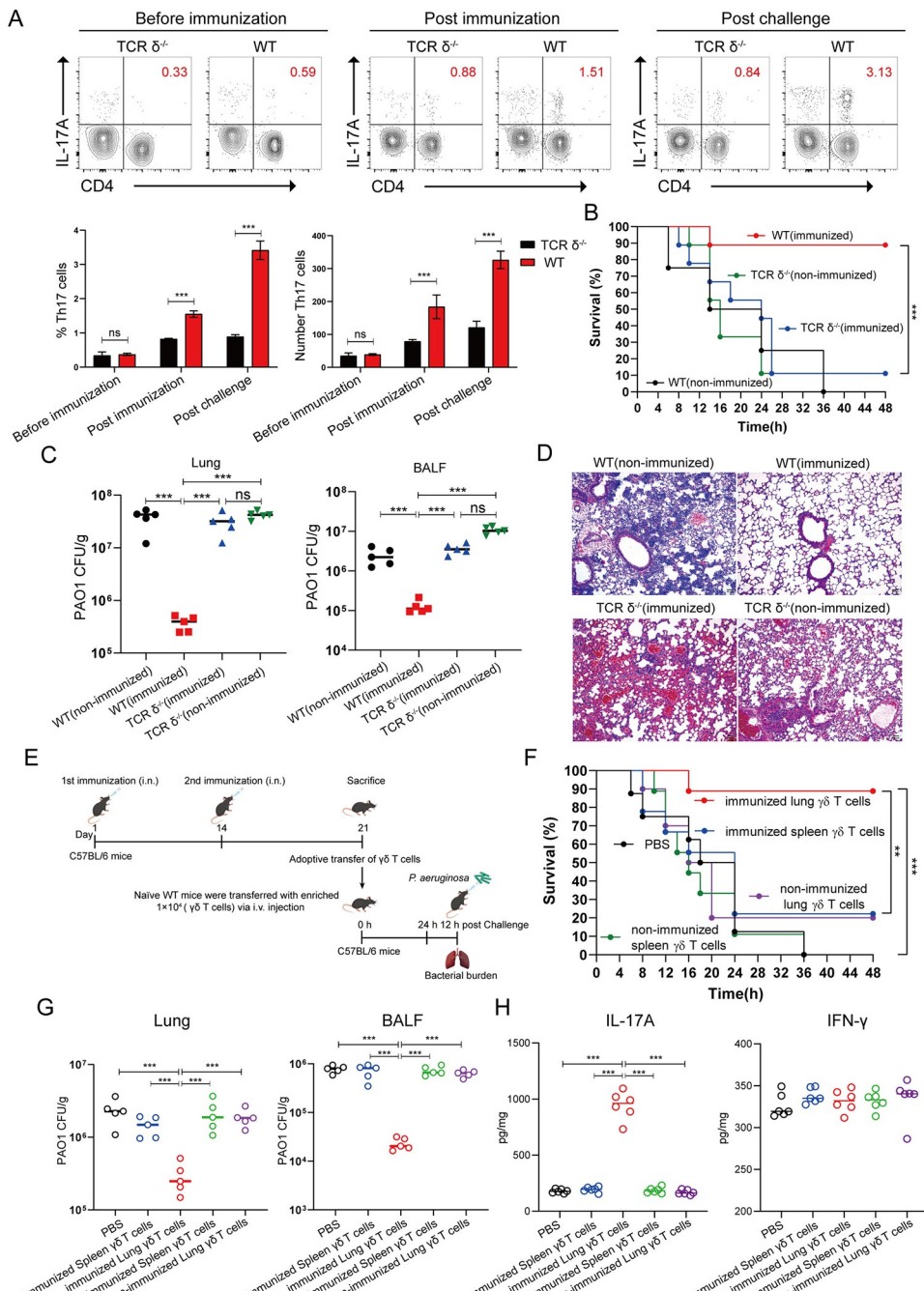

**Fig 6. γδ T cells mediate immune response induced by the EPS301-adjuvanted vaccination against *P. aeruginosa* pneumonia.** TCR δ-deficient mice were vaccinated with EPS301@rPcrV and survival, bacterial loads and the lung pathology were detected on day 7 post second vaccination. Mice were inoculated with 40 μl bacterial slurry (1×10⁹ CFUs of *P. aeruginosa* PAO1) via intranasal route (i.n.). (A) The number and percentage of Th 17 cells in WT and TCR δ-deficient mice before immunization, after immunization and post infection. (B) Survival of mice were observed for 48 h. Mice were inoculated with 40 μl bacterial slurry (1×10⁷ CFUs of *P. aeruginosa* PAO1) via intranasal route (i. n.). The data for survival test were analyzed by Wilcoxon log-rank survival test (***p < 0.001). (C) Bacterial burdens of mice were counted at 12 h post infection. (D) Histological evaluation of lung sections by light microscopy. Lung specimens were fixed, sectioned, and stained with H&E (n = 3–5). (E) Timeline of adoptive transfer and *P. aeruginosa*-induced pneumonia model. Lung and splenic CD3⁺γδ T cells from EPS301@rPcrV–vaccinated congenic mice were purified on day 7 after second vaccination. CD3⁺γδ T cells (1 × 10⁴) were intravenously transferred into naïve C57BL/6 mice. (F) 24 hours after adoptive transfer, mice were inoculated with 40 μl bacterial slurry (1×10⁹ CFUs of *P.*

*aeruginosa* PAO1) via intranasal route (i.n.). Survival of mice were observed for 48 h. Survival after *P. aeruginosa* challenge (data were pooled from two independent experiments (n = 8–10). The data for survival test were analyzed by Wilcoxon log-rank survival test (**p < 0.01, ***p < 0.001). A lower dose of bacterial slurry (1×10^7 CFUs of *P. aeruginosa* PAO1) was inoculated to recipient mice via i.n.. (G) Bacterial loads in lung and BALF were detected at 12 h post infection. (H) 12 hours post-challenge, lung tissue was prepared. IL-17A and IFN-γ levels determined by ELISA in a supernatant of lung tissue homogenate were analyzed. Data are presented as means ± SEM. Significant differences were calculated with unpaired t test (A) and One-way ANOVA followed by Tukey's multiple comparisons test (C, G and H). ns, not significant, *p < 0.05, **p < 0.01, ***p < 0.001.

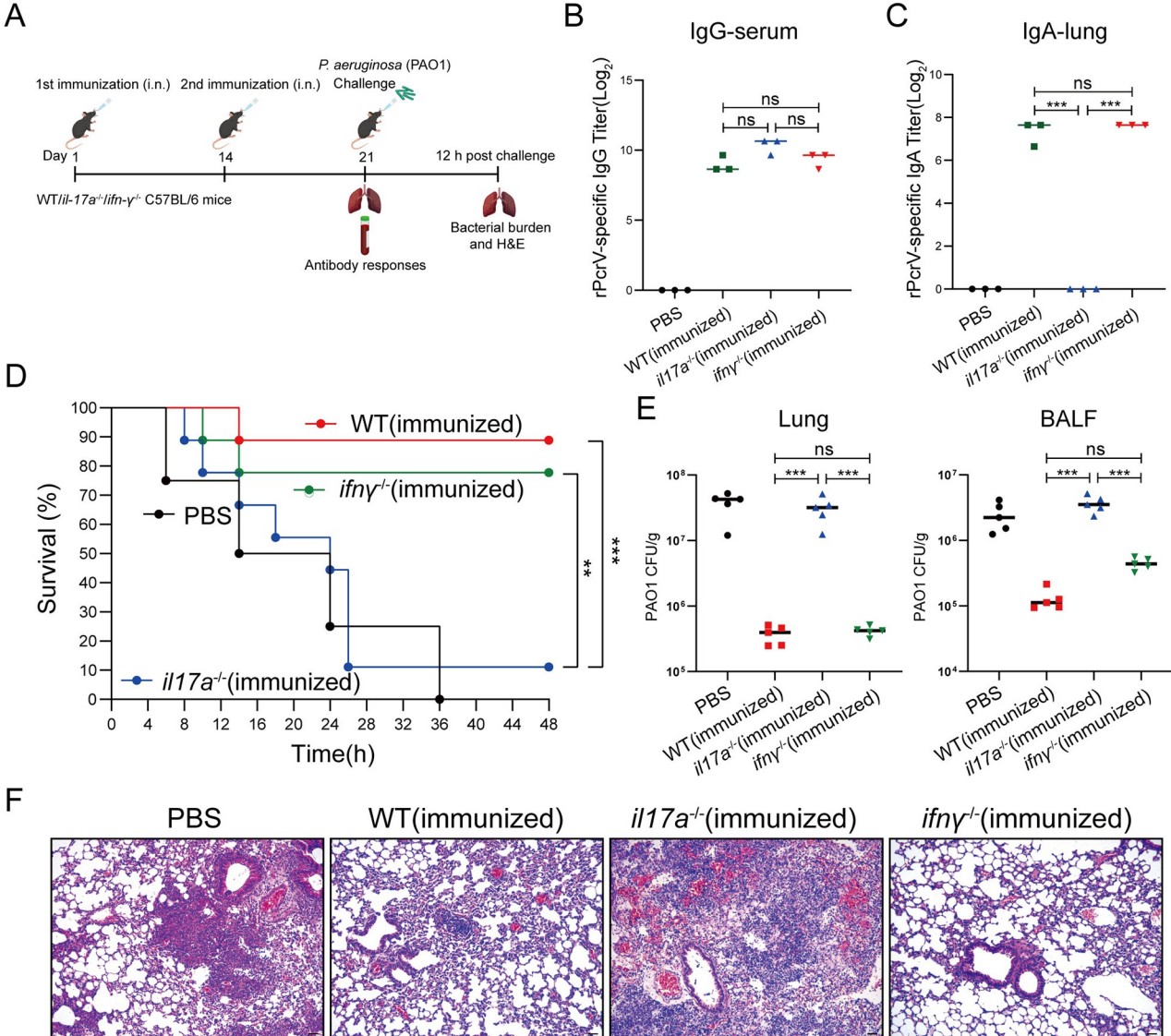

**Fig 7. EPS301-adjuvanted vaccination was dependent on IL-17A induced immunity responses.** (A) Experimental design and timeline of prime and booster regimen to IL-17A-deficient, IFN-γ-deficient and WT mice of EPS301-adjuvanted vaccination (i.n.) and challenge. (B) 7 days post booster vaccination, level of IgG in serum was detected by ELISA. (C) 7 days post booster vaccination, level of IgA in lung was detected by ELISA. (D) IL-17A-deficient, IFN-γ-deficient and WT mice were inoculated with 40 μl bacterial slurry (1×10^9 CFUs of *P. aeruginosa* PAO1) via left nostril on day 7 post booster vaccination and held upright for 1 min. Representative survival rates from two independent experiments are shown (n = 8–10). The data for survival test were analyzed by Wilcoxon log-rank survival test (**p < 0.01, ***p < 0.001). (E) Mice were inoculated with 40 μl bacterial slurry (lower dose: 1×10^7 CFUs of *P. aeruginosa* PAO1) as above. At 12 h, the numbers of bacteria in lung and BALF were counted (n = 5–8). (F) Mice were inoculated with the lower dose of PAO1 as above. At 12 h post challenge, histological evaluation of lung sections by light microscopy. Lung specimens were fixed, sectioned, and stained with H&E (n = 3–5). Data are presented as means ± SEM. Significant differences were calculated with One-way ANOVA followed by Tukey's multiple comparisons test. ns, not significant, *p < 0.05, **p < 0.01, ***p < 0.001.

showed higher bacterial loads in the lungs and BALF compared to vaccinated WT and IFN-γ-deficient mice (Fig 7E). Then, we examined the stained lung tissues. The lung tissues of EPS301-adjuvanted vaccinated WT or IFN-γ-deficient mice showed clear alveolar airspaces and clear delineation of extra-alveolar vessels. Pneumonocytes were intact, and cell structures were clear with inflammatory cell infiltration. In contrast, lung tissues of non-vaccinated or vaccinated IL-17A-deficient mice displayed interstitial edema, alveolar wall thickening and notable damage to pneumonocyte morphology (Fig 7F). These results demonstrated that EPS301-adjuvanted vaccination was dependent on IL-17A induced immunity responses.

## Discussion

Mucosal infections caused by *P. aeruginosa*, including respiratory infection and ocular infection are a major health problem globally [8,42]. Vaccination is an effective approach for preventing *P. aeruginosa* infection. However, despite substantial research efforts over the past fifty years, a vaccine licensed for clinical use has not yet been approved, and several challenges remain to be addressed [7,11]. The induction of protective immunity in humans remains challenging and identifying safe yet potent mucosal adjuvants to enhance local protective immune responses remains a critical roadblock for the development of an effective *P. aeruginosa* vaccine.

Mucosal vaccination is a highly effective and recommended method to prevent mucosal-transmitted infections. Compared with immunization via intramuscular injection, mucosal immunization offers remarkable advantages, including non-invasiveness, low costs, and reduced risk of transmission of blood-borne diseases [43–44]. Despite all of these advantages, there are not many mucosal vaccines approved for humans, which is mainly due to the deficiency of safe and effective mucosal adjuvants [45]. Traditional vaccines are applied parenterally with adjuvants meant to induce a powerful serotype-dependent response which often fail to drive mucosal immune protection. Alum, the current gold-standard in vaccine adjuvants is known to elicit a Th2 response, whereas, *P. aeruginosa* require the different type of cellular immunity [46–47]. Several mucosal adjuvants, such as cholera toxin (CT) or Escherichia coli heatlabile toxin (LT) have been found to be effective in conferring protection against bacterial infections. However, the toxicity of these potent enterotoxins precludes their use as mucosal vaccine adjuvants in humans [48–49]. Additionally, CT has been proved to cause excessive inflammation. These effects may not be immediately apparent after vaccination, but become evident during infection when the immune system is activated, witch, in part, explained why the mice in the CT-adjuvant immunization group in this study experienced more severe lung damage after *P. aeruginosa* infection [50]. Many studies are currently exploring effective adjuvants and vaccine delivery vehicles for mucosal vaccines, which can enhance the immune protection induced by mucosal vaccines with different mechanisms of action [51].

The applications of polymers as carrier systems have flourished in the delivery of mucosal vaccines because they provide the advantage of delivering antigens to specific targets. Another benefit of polymer carrier systems is that they control the way antigens are released slowly or by burst release at their mucosal sites [52]. Due to the great potential of synthetic and natural polymers and their desired physical and chemical properties, these polymer materials are used as the most appreciated option for developing nanoparticles for mucosal vaccine delivery systems. Microbial exopolysaccharides (EPS) are polymers of high molecular weight carbohydrates composed of homo- or hetero-monosaccharides. Due to their unique and complex chemical structures that offer beneficial bioactive functions, biocompatibility, and biodegradability, microbial EPSs have been found a wide range of application areas in chemical, food, pharmaceutical, cosmetics, packaging industries, agriculture, and medicine [53]. Most of the microbial EPS have irregular structures, such as porous [54], feather-like [55] and sheet-like

structures [56]. Furthermore, it has been reported that EPSs can appear at the nanoscale in different conformations in solution, such as spherical nanoparticles and nanofibers, conferring them multiple roles [57–58]. Xu et al [59]. reported that fungus produce an EPS composed of β-glucan, which can self-assemble into hollow nanofibers with a diameter of fewer than 100 nanometers and a length of tens of micrometers in a diluted solution. Zhao et al [60]. found that *Klebsiel sp* EPS of PHRC1.001 exists in the form of nanoparticles (about 50 nm in diameter) in an aqueous solution, and has a strong tendency to aggregate. Robert et al [61], reported the isolation of an exopolysaccharide (EPS)-producing Lab strain KM01 from Thai fermented desserts, which formed large aggregates (MW > 2000 kDA) in aqueous solution. In addition, the formation of hydrogels and nanoparticles of KM01 EPS was found to be reversible. Li et al [62]. observed that a novel non-glucan EPS named EPS-605 can self-assemble to form spherical nanosize particles of ~88 nm in diameter, and so on. So far, only β-glucans have been reported for the self-assembly of EPSs, and only two self-assembled structures (nanofibers and triple helices) have been reported for these EPSs [63–64]. It would certainly be worthwhile to identify novel EPSs that can self-assemble into nanoparticles in solution.

In this study, we utilized the self-assembling EPS301 to encapsulate *P. aeruginosa* antigen and formulate a nano-vaccine. Fluorescent imaging experiments indicated that, compared to immunization with rPcrV alone, EPS301@rPcrV prolonged the retention time of the antigen in the nasal mucosa. Six hours after intranasal immunization, the fluorescent signal was observed to undergo secondary move to the lungs. We speculate one of the potential reasons is that the drying of the instilled EPS301@rPcrV in the nose, followed by the inhalation of crusts during breathing, leading to the secondary move of the fluorescent signal. Another reason could be that EPS301 acts as a mucosal adjuvant specifically targeting the lungs, promoting antigen accumulation in the lungs and consequently causing the secondary transfer of the fluorescent signal. Nonetheless, the precise mechanism remains unclear. EPS301-adjuvanted nano-vaccine provided effective and sustained protective immune responses against *P. aeruginosa* induced pneumonia even superior to the mucosal adjuvant "gold standard", CTB. However, we found that while EPS301 demonstrated better performance than the CTB adjuvant in inducing sustained protection, the survival rate of mice infected with *P. aeruginosa* 112 days after immunization significantly decreased to around 60%, compared to the survival rate 7 days after immunization. This indicates that further research on the next generation of adjuvants should focus on improving the durability of immunity.

Historically, vaccine development has focused on driving antibody responses. Antibodies to *P. aeruginosa* are important in blocking toxins involved in lysing immune cells, as well as providing opsonic help to phagocytes. However, the clinical trials using a fused protein (OprF/I)-based vaccine candidates failed to show efficacy in protection against pneumonia, which may be, in part, due to an overreliance on an antibody-mediated protective response [37,65]. Herein, given that EPS301-adjuvanted vaccine formulation elicited strong and long-term antigen-specific pulmonary IgA and systemic IgG responses. However, serum from immunized mice promoted opsonic macrophage activity but is insufficient to protect mice from *P. aeruginosa* pneumonia. SenKilic et al. recently highlighted the importance of the humoral immune response from a curdlan-adjuvanted *P. aeruginosa* whole-cell vaccine (WCV) by passive immunization [66]. Wei Sun et al. investigated an OMVs-delivery PcrV-HitAT vaccine to *P. aeruginosa* and proved that it is T cells, but not systemic antibodies, are essential for protection against pulmonary *P. aeruginosa* infection [9]. Additionally, a previous report demonstrated that vaccination against PcrV ensured the survival of challenged mice and antibodies to PcrV inhibited the translocation of type III toxins [67]. The discrepancies in these results may stem from differences in vaccine type, *P. aeruginosa* challenge strain, or mouse strain used across the studies. Our opsonophagocytosis assay suggests that transferring larger volumes of anti-

rPcrV serum from immunized mice may offer partial protection. Therefore, further investigation is necessary to elucidate the specific roles of B and T cells induced by EPS301@rPcrV immunization in mediating protection.

The contribution of Th17 immunity to the prevention of infection by pulmonary pathogens has gradually been recognized [68]. Pulmonary Th17 cells participate in the recruitment of neutrophils, the release of antimicrobial peptides, IL-17–driven Th1 immunity and so on [69]. These effectors provide immunity against a wide range of pathogens through the antibody independent pathway. In addition, Th17 cells have been shown to be crucial for mucosal antigen-specific IgA responses [70]. Therefore, inducing protective Th17 responses is of great importance when designing a vaccine against a pulmonary pathogen. The Th17 response is also protective against pulmonary *P. aeruginosa* infection. To date, the number of Th17 cells and the concentration of IL-17A in the lungs have been shown to increase significantly as soon as 4 h after *P. aeruginosa* infection [71]. Blocking the Th17 response leads to more severe pathological damage in the lung [35]. We, therefore, detected the short- and long-term cellular immune responses induced by EPS301-adjuvanted vaccination. Both Th17 and γδ17 T immune responses in lung, as well as in spleen were significantly enhanced by EPS301-adjuvanted vaccination. Adoptive transfer results demonstrated EPS301-adjuvanted vaccination immunized lung CD4 T cells and γδ T cells were crucial for anti-*P. aeruginosa* induced pneumonia, but immunized spleen CD4 T cells and γδ T cells were dispensable, which, in part, explained the traditional parenteral vaccine were failed to *P. aeruginosa* pneumonia because of mainly induced systemic immune response. Several papers have shown that lung TRM cells generated from Th 17 cells can provide prompt protection to extracellular pathogens [38–39]. We proved that both CD4 and γδ T cells induced by EPS301 expressed CD44 and CD69. Vaccinated mice treated with FTY720 maintained protection against *P. aeruginosa*, which aligns with findings that only transferring pulmonary CD4 or γδ T cells is sufficient to control pulmonary bacterial infection. Our findings from the acute *P. aeruginosa* pneumonia model may not directly translate to chronic *P. aeruginosa* infection in cystic fibrosis (CF) patients. Importantly, individuals with CF exhibit permanently elevated levels of IL-17 in their sputum and bronchoalveolar lavage fluid, alongside infiltration of Th17 lymphocytes in the airway submucosa [72]. While these observations do not exclude the potential effectiveness of Th17 responses prior to infection establishment, it is plausible that vaccine-induced Th17 responses may prove ineffective in the CF lung and could potentially exacerbate neutrophilic airway inflammation associated with CF.

In summary, we demonstrated that the intranasal EPS301-adjuvanted vaccination provided effective and sustained protection against *P. aeruginosa* induced pneumonia in an IL-17A–dependent manner. Importantly in comparison with other known assembly strategies such as enzymatic self-assembly and chemical assembly [73–74]. EPS-based self-assembly can avoid the supplement of enzymes or chemical catalysts, therefore simplifying the formulation process of vaccines. The satisfactory biocompatibility and biosafety of EPS301 further facilitates its clinical applications. Taken together, all the above advantages highlight the promising potential of EPS301 for a final transformation and application in the field of the vaccine. Additionally, our findings provided direct evidence that EPS301@rPcrV mucosal vaccine is a promising candidate for future clinical application against *P. aeruginosa*-induced pulmonary infection.

## Materials and methods

### Ethics statement

The study was conducted according to the guidelines of the Declaration of Helsinki. All animal related experimental protocols applied in this study were conducted under the standards of Ethics Committee of Inner Mongolia University (IMU-2022-mouse-054).

## Experimental animals

Specific-pathogen-free (SPF) C57BL/6 mice and BALB/c mice, age 6–8 wk, were purchased from Beijing Vital River Laboratory Animal Technology Co. (Beijing, China). IFN-deficient mice (JAX Stock number: 002116), γδ T cell-deficient mice (JAX Stock number: 003288) and IL-17A-deficient mice (NCBI Stock number: 16171) on C57BL/6N background were kindly donated by Dr. Z. Yin (College of Life Sciences, Jinan University, Guangzhou, China). All animals were housed for 7 days to adjust to housing conditions under a strict 12 h light/dark cycle and fed ad libitum. Animals were euthanized by cervical dislocation following administration of ketamine/xylazine cocktail at defined time-points post-infection (see Figure legends) or when humane endpoints (e.g. hypothermia, weight loss) had been reached, whichever occurred earlier.

## Bacterial strains

*Lactobacillus plantarum* WXD301 (*L. plantarum* WXD301) was isolated from lung commensal microbiota from C57BL/6 WT mice in Inner Mongolia, China, and maintained in our laboratory. It was identified as *L. plantarum* on the basis of physiological characteristics and 16S rDNA sequence analysis.

   *P. aeruginosa* strains PAO1 was purchased from BeNa Culture Collection (Beijing, China).

## Preparation of EPS301 based nanoparticles

The homogeneity and molecular weight of EPS were determined by high-performance gel permeation chromatography (HPGPC, Shimadzu LC-10AD, Kyoto, Japan), with BRT105-104-102 column (8 mm×300 mm, 35 ˚C) in series and a differential refractive index detector (RID-10A). For monosaccharide composition analysis, EPS301 (10mg) was hydrolyzed with trifluoroacetic acid (3 M, 10 mL) at 120 ˚C for 3 h and dried by vacuum rotary evaporation. The hydrolysates were dried under a stream of $N_2$ and dissolved with 10 mL ultrapure water. The IC (ion chromatography) system used was a Dionex ICS3000 equipped with a conductivity detector and a DionexCarboPacTMPA20 analytical column (3mm×150 mm) and running at a flow rate of 0.3 ml/min at 30˚C. The monosaccharide standards (fucose, galactosamine hydrochloride, rhamnose, arabinose, glucosamine hydrochloride, galactose, glucose, N-acetyl-D glucosamine, xylose, mannose, fructose, ribose, galacturonic acid, guluronic acid, glucuronic acid, mannuronic acid) were processed with the same procedure.

   Production and purification of exopolysaccharide was described previously [32]. To study the self-assembly characteristics of EPS301, a series of treatments were performed on EPS301. Briefly, the purified exopolysaccharide was dissolved in deionized (DI) water at 1 mg/ml. Then the solution was stirring at room temperature for 2 h following lyophilizing to collection. The morphological features of exopolysaccharides were recorded using a scanning electron microscope (SEM, S-4800, Tokyo, Japan). Transmission electron microscope (TEM, HITACHI, HT-7800, Japan) was performed to observe the morphology of the nanoparticles. Average particle size and Zeta potential of exopolysaccharides were determined by the dynamic light scattering instrument (DLS, NanoBrook, 90Plus, PALS, U.S). The EPS301@rPcrV was prepared by homogenization methods. In brief, 10 mg of EPS301 was dissolved in 10 ml of DI water at 200 rpm stirring for 2 h, 10 mg of rPcrV was dissolved in 10 ml of DI water. Subsequently, the rPcrV solution was mixed with the EPS301 solution at a 1:1 ratio and equilibrated at room temperature for two hours, followed by dialysis in DI water using a 10,0000 MWCO (molecular weight cut-off) membrane at 4 ˚C for two days. Afterward, the solution of EPS301 encapsulation was concentrated using a 50,000 MWCO membrane. Further quantify EPS301@rPcrV using the phenol-sulfuric acid method and adjust the final concentration of EPS301 to 10 mg/

ml in DI water. Then, EPS301@rPcrV was lyophilized for SEM detection. Adjusted the final concentration of EPS301 to 1 mg/ml with DI water for TEM, DLS and Zeta potentials analysis.

## In vivo imaging

In order to figure out the pharmacokinetics, fluorescent in vivo imaging was carried out. The rPcrV or OVA was first labeled with fluorescent dye Cy7 (Bioss, BA00122). Cy7-rPcrV or Cy7-OVA was used to prepare EPS301@Cy7-rPcrV or EPS301@Cy7-OVA nanoparticles as described above. Then BALB/c mice were treated with Cy7-rPcrV, EPS301@Cy7-rPcrV, Cy7-OVA and EPS301@Cy7-OVA by intranasal route, respectively. Mice were anesthetized by isoflurane and imaged by a fluorescence vivo imaging system (VISQUE, Smart-LF, Kibero) at different time points (5mins, 3 h, 6 h, 9 h, 12 h, 24 h, 48 h, 72 h and 96 h) after inoculation. The images were then analyzed by Fiji software to compare the fluorescence intensity of these groups.

## Expression analysis of recombinant PcrV

*pcr*V gene sequence (885 bp) was retrieved from GenBank database, and amplified by polymerase chain reaction (PCR) from genomic DNA of *P. aeruginosa* strains ATCC27853 using primers F: 5′- ATGGAAGTCAGAAACCTTAATGC -3′/ R: 5′-CTAGATCGCGCTGA-GAATGT-3′ (S1A Fig). PCR products were cloned into pEASY-Blunt E1 expression vector with N-terminal 6-histidine-tag (His-tag), and transformed into *Escherichia coli* BL21(DE3) (TransGen Biotech, Beijing). Protein expression was induced by adding isopropyl β-D-1-thio-galactopyranoside (IPTG) to final concentration 1 mM, and cultures were grown at 37 ˚C with shaking (200 rpm) for 12 h. rPcrV antigen expression was evaluated *in vitro* by Western blotting analysis (S1B Fig).

Cells were harvested by centrifugation at $8000 \times g$ for 15 min at 4 ˚C. After resuspension and ultrasonication, the supernatant was carefully collected and purified by Ni-NTA affinity chromatography (TransGen Biotech). Expressed rPcrV protein was estimated as 31 kDa on the basis of SDS-PAGE analysis and gel filtration chromatography (S1C Fig).

## Vaccination

As described above, lyophilized EPS301, rPcrV, CTB+rPcrV and EPS301@rPcrV dissolved with phosphate-buffered saline (PBS) and stored at 4 ˚C. Wild-type (WT), IFN-γ-deficient (*ifnγ*$^{-/-}$), IL-17A-deficient (*il17a*$^{-/-}$), and γδ T cell-deficient mice (TCR δ$^{-/-}$) were injected with 180 μl 4% chloral hydrate. Mice were then anesthetized and vaccinated by intranasal (i.n.) route (40 μl volume per mice). Each mouse received a dose corresponding to 40 μg EPS301 and 40 μg rPcrV both two vaccinations. Two vaccinations were given at 2-wk interval.

## Murine pneumonia model

For the short-term studies, animals were challenged on day 7 after the last dose of the vaccine administration. For the long-term study, animals were challenged 112 days after the last dose of the vaccine deliver. Vaccinated mice were injected with 180 μl 4% chloral hydrate and then anesthetized, inoculated with 40 μl bacterial slurry ($1\times10^9$CFUs of *P. aeruginosa* PAO1) via left nostril, and observed continuously for 48 h-72 h. 40 μl bacterial slurry ($1\times10^7$ CFUs of *P. aeruginosa* PAO1) was inoculated via left nostril, and numbers of bacteria in organs at 12 h post challenge were counted by preparing organ homogenates in PBS and plating on tryptic soy agar (BD Diagnosis System). Colonies were counted after 24 h incubation at 37 ˚C. Left lobes

of lung were removed at 12 h post challenge, and tissues were fixed, paraffin-embedded, sectioned (4–6 μm), and stained with hematoxylin and eosin (H&E).

## ELISA for specific antibodies

Antibody titers were determined by using indirect Enzyme-linked immunosorbent assay (ELISA). Briefly, ELISA plates were coated with 5 ng/μl of purified rPcrV and allowed to adhere to the plates overnight at 4 ˚C. Antigen coated plates were then blocked with 1% powered milk-PBS solution for 1 h at room temperature. Minimum dilution was 1:100, and then two-fold serial dilutions of lung homogenate or serum was added to the plates and incubated for 1 h at room temperature. After 3 times washes with a 0.05% Tween 20 in PBS solution, horseradish peroxidase (HRP)-conjugated goat anti-mouse IgG (1:5000 dilution) or IgA (1:500 dilution) antibodies (Proteintech, Beijing China) were then applied for 1 h at room temperature. After three times washes, the reaction in the plates was developed using 3,3',5,5'-Tetramethylbenzidine (TMB) solution (Sigma–Aldrich, St. Louis, MO) (100 μl/well) for 5–15 min at room temperature. The reaction was stopped using 2% $H_2SO_4$ (50 μl/well). Color development was read on a Versamax tunable plate reader (Molecular Devices San Jose, CA) at 450 nm. IgG and IgA were measured at least in 3 replicates. The OD value was used for data involving extremely high titers, and for graphical presentation of data.

## Opsonophagocytosis assay

Murine macrophage cell line RAW264.7 (TIB-71) was cultured Dulbecco's modified Eagle's medium (DMEM) (Gibco, 1195500BT) and supplemented with 10% HI fetal bovine serum (Gibco, 10099141C), antibiotics (penicillin and streptomycin, Gibco, 15140122), sodium pyruvate (Gibco, 11360070), and non-essential amino acid (Gibco, 11140050), and grown at 37 ˚C under an atmosphere of 5% $CO_2$. Serum samples from immunized mice (7 days post second immunization) containing opsonic antibodies were heat inactivated (56 ˚C, 30 min) and serially diluted with opsonization buffer (mixture of 80 ml of sterile water, 10 ml of 10×Hanks balanced solution, 5 ml of 2% gelatin, and fetal bovine serum to the final concentration of fetal bovine serum at 5%). Each well in a 96-well plate contained 40 μl of $5×10^5$ RAW264.7 cells, $1×10^5$ CFU, $1×10^4$ CFU of *P. aeruginosa* PAO1 in 10 μl of opsonophagocytic buffer, 20 μl of serum, and 10 μl of 1% infant rabbit serum as a complement source. The infected cells were washed twice with HBSS and lysed with 0.2% Triton X-100 (Sigma, X100) after 3 h incubation. The lysate was serially diluted and plated onto Luria-Bertani agar plates for CFU enumeration. Each experimental group was assayed in triplicate, and three independent experiments were performed.

## Passive transfer

Animals immunized with rPcrV, CTB+rPcrV and EPS301@rPcrV were euthanized on day 7 after the booster vaccination. 100 μl of pooled sera from donor mice were transferred to recipient female C57BL/6 mice (6–8-week-old) (n = 9–10/group) by i.v. injection. 24 hours after serum transfer, Antigen-specific IgG titers of recipient mice were determined by using indirect Enzyme-linked immunosorbent assay (ELISA). And then, recipient mice were challenged with 40 μl bacterial slurry ($1×10^9$ CFUs of *P. aeruginosa* PAO1) via the i.n. route. The survivals of mice were monitored and recorded for 48 h-72 h after challenge.

## ELISA for cytokine

Seven days or 112 days after the booster vaccination, mice (n = 3-5/group) were challenged with the lower dose of *P. aeruginosa* PAO1 (challenge dose, $1×10^7$ CFUs of *P. aeruginosa*

PAO1). At 12 h post infection, mice were sacrificed and the lung and spleen were harvested. The lung and spleen (0.1 g tissue into 100 μl PBS) were homogenized using tissue grinders (High-throughput Tissue Grinders NANBEI, NB-48P) and centrifuged at 800×g at 4 ˚C for 10 min, then supernatant was collected and stored at −80 ˚C. To define the IL-17A and IFN-γ levels in lung and spleen tissue homogenate, the murine bioplex ELISA kits (R&D SYSTEMS Mouse IL-17A/F Heterodimer DuoSet ELISA DY5390 and R&D SYSTEMS Mouse IFN-gamma DuoSet ELISA DY485) were used according to the manufacturer's recommendations.

## Flow cytometric analysis

For analyzing the changes of T cells post immunization and after challenge, animals (3–5 mice per group) were sacrificed at 12 h after challenge (challenge dose, 1×10^7 CFUs of *P. aeruginosa* PAO1) on day 7 and day 112 after booster vaccination and lung and spleen cells were prepared. For spleen, single-cell suspensions were isolated through a 100 μm nylon cell strainer (BD Nylon, 352340) then treated with 1× Red Blood Cell lysis buffer (TIANGEN). Splenocytes were adjusted to a concentration of 1.5×10^6 cells/ml (10:1). For lung cell isolation, organs were transferred to a petri dish and disrupted with sharp scissors prior to passage through a 100 μm nylon cell strainer (BD Nylon, 352340). Lung cell suspensions were centrifuged twice at $60 \times g$ for 1 min at room temperature (RT), to remove fibroblasts and other large non-leukocytes, then the supernatant was transferred to a new tube. The supernatant was centrifuged at $300 \times g$ for 5 min and cell pellets resuspended in media. Lung cells were seeded 1.5×10^5 cells/ml (1:1). Single-cell suspension from mouse lungs and spleens were blocked with αCD32/CD16 (BioLegend, purified CD16/CD32 monoclonal antibodies can bind to FcγRIII/II, blocking nonspecific staining) at a dilution of 1/200 and then stimulated for 6 hours with phorbol 12-myristate 13-acetate (50 ng/ml; Sigma-Aldrich) and ionomycin (750 ng/ml; Sigma-Aldrich). After 1 hour of incubation, GolgiStop (1 mg/ml; BD Biosciences) was added to block cytokine secretion. Cells were then stained with Fc block (BD Bioscience, 553142) to reduce non-specific binding of conjugated antibodies at 4 ˚C for 30 min. Surface staining with anti-CD45 (eBioscience PerCP-Cyanine5.5 45-0454-82), anti-CD3 (eBioscience APC 17-0031-82), anti-CD4 (eBioscience FITC 11-0041-82), anti-TCR γ/δ (eBioscience PE-Cyanine7 25-5711-82), anti-CD69 (eBioscience PE 12-0691-83), anti-CD44 (BD Biosciences APC-Cyanine7 2306091) antibody at 1:100 dilution was performed for 30 min at 4 ˚C, and data were analyzed using De Novo software program. For study intracellular cytokines, briefly, cells were washed with PBS, and subsequently fixed and permeabilized using the Fixation/Permeablization Kit (BD Biosciences, 554714) as directed by the manufacturer. Following fixation, cells were washed in Permwash buffer (BD Biosciences, 554714) and incubated with intracellular molecule IL-17A (eBioscience PE 12-7177-81) or IFN-γ (Biolegend PE B240651) for 1 h. Cells were washed and fixed in 4% ultra-pure formaldehyde (Cell Signaling, 477465) for 48 h and finally placed in 2% formaldehyde prior to analysis by flow cytometry (ACEA NovoCyte).

## FTY720 treatment

Fingolimod hydrochloride (FTY720, MedChemExpress, 162359-56-0) was initially dissolved in dimethyl sulfoxide (DMSO) to create a 50 mg/mL stock solution. This stock solution was subsequently diluted with saline for intraperitoneal administration at a dosage of 30 μg per mice, and with distilled water for administration via stomach intubation at a dosage of 50 μg per mice.

## Adoptive transfer of CD4$^+$ T cells or γδ T cells

Animals immunized with EPS301@rPcrV were euthanized on day 7 after the booster vaccination. The spleens and lungs of immunized mice were harvested and homogenized. The single suspension of spleen and lung cells was obtained by manually pushing spleen homogeneity through a cell strainer (40-μm pore size). Cells were enumerated and resuspended in PBS containing 0.5% bovine serum albumin (BSA) and 2 mM EDTA. γδ cells were isolated using a TCR γ/δ isolation kit for mice (Miltenyi Biotec 130-092-125) according to the manufacturer's instructions. CD4 T cells were isolated using a CD4 isolation kit for mice (Miltenyi Biotec 130-104-454) according to the manufacturer's instructions.

A total of 100,00 γδ T cells or 50,000 CD4 T cells in the flowthrough were transferred to mice through intravenous injection at 24 h prior to *P. aeruginosa* pneumonia inoculation.

## Statistical analysis

GraphPad software program was used for data processing and analysis. P values < 0.05 were considered statistically significant. Comparisons between two normally distributed groups were performed by simple two-tailed unpaired Student's t test, and other analyses for unequal variance were determined by Mann-Whitney U test. For multiple groups comparisons, we used One- or Two- way analysis of variance (ANOVA) with Tukey's post hoc analysis for equally distributed groups. The data for survival test were analyzed by Wilcoxon log-rank survival test. Values are presented as means ± SEM. P values are annotated as follows: *p < 0.05, **p < 0.01, ***p < 0.001.

## Supporting information

**S1 Fig. Cloning, expression, purification, and evaluation of recombinant PcrV.** (A) pcrV gene sequence (885 bp) was was cloned from *P. aeruginosa* PAO1. Arrows: *pcrV* gene. (B) rPcrV expression was evaluated in vitro by Western blotting analysis. Lane M, standard size marker (kDa); lane1 and 2 cell lysis protein of BL21(pEASY-Blunt E1- pcrV); lane3 and 4 cell lysis protein of BL21(pEASY-Blunt E1). Arrows: rPcrV protein. (C) Samples were resuspended in SDS loading buffer and boiled for 5 min. Lane M, standard size marker (kDa); lane 1, pellet of non-induced bacteria; lane 2, pellet of IPTG-induced bacteria; lane 3, purified rPcrV from Ni-NTA agarose column. Gel was stained with Coomassie Brilliant Blue R-250. Arrows: rPcrV protein.
(TIF)

**S2 Fig. Characteristics of EPS301.** (A) Photo of lyophilized EPS301. (B) Photo of EPS301 aqueous solution. (C) The homogeneity and molecular weight of EPS301. (D) The monosaccharide composition of EPS301. (E) The scanning electron microscopy (SEM) images of EPS301 and EPS301@rPcrV. (F) The solution (1.0 mg/mL) of EPS301 in PBS was prepared, whose particle size, of EPS301 were measured by the dynamic light scattering (DLS) technique using a Nano Brook (Nano Brook, 90 Plus, PALS, U.S). The values presented are the average of three measurements, and the standard deviation was considered as the error range. (G) The solution (1.0 mg/mL) of EPS301 in PBS, serum and 1640 medium were prepared, whose particle size, of EPS301@rPcrV were measured by the dynamic light scattering (DLS) technique using a Nano Brook (Nano Brook, 90 Plus, PALS, U.S). The values presented are the average of three measurements, and the standard deviation was considered as the error range. (H) Representative in vivo fluorescence images of mice at the indicated time points after intranasal administration of free OVA or EPS301@OVA. OVA in both groups labeled with Cy7. Significant differences were calculated with One-way ANOVA followed by Tukey's multiple

comparisons test. ns, not significant, **p < 0.01, ***p < 0.001. Data are presented as means ± SEM.
(TIF)

**S3 Fig. Assessing the safety of EPS301.** Histological evaluations conducted before challenge. Mice were immunized twice via airway (intranasal, i.n.) administration routes. Animals (3–5 mice per group) were sacrificed on day 7 after boost immunization for histological evaluation of lung sections using light microscopy. Lung specimens were fixed, sectioned, and stained with H&E (n = 3–5).
(TIF)

**S4 Fig. Changes of CD8$^+$ T cells in vaccinated mice post challenge.** Mice (n = 3-5/group) were immunized (i.n.) twice 14 days apart with rPcrV, CTB+rPcrV or EPS301@rPcrV, with animals receiving PBS served as controls. Vaccinated mice were sacrificed at 12 hours post-challenge on day 7 after the second vaccination lung tissue were prepared. Number of CD8$^+$ T cells were determined by Flow cytometric. Data are presented as means ± SEM. Significant differences were calculated with One-way ANOVA followed by Tukey's multiple comparisons test. ns, not significant.
(TIF)

**S5 Fig. Intranasal immunization adjuvanted with EPS301 provoked superior cellular immune responses.** Mice (n = 3-5/group) were immunized (i.n.) twice 14 days apart with rPcrV, CTB+ rPcrV or EPS301@rPcrV, with animals receiving PBS served as controls. Mice were inoculated with 40 μl bacterial slurry (lower dose, 1×10$^7$ CFUs of *P. aeruginosa* PAO1) as above. Vaccinated mice were sacrificed at 12 hours post-challenge on day 7 or day 112 after the second vaccination spleen tissue were prepared. Number of IFN-γ$^+$ CD4$^+$ T cells, IL-17A$^+$ CD4$^+$ T cells (A), IFN-γ$^+$ γδ$^+$ T cells, IL-17A$^+$ γδ$^+$ T cells (B) in lung at 12 hours post-challenge on day 7 after the second vaccination were estimated by intracellular cytokine. Number of IFN-γ$^+$ CD4$^+$ T cells, IL-17A$^+$ CD4$^+$ T cells (C), IFN-γ$^+$ γδ$^+$ T cells, IL-17A$^+$ γδ$^+$ T cells (D) in lung at 12 hours post-challenge on day 112 after the second vaccination were estimated by intracellular cytokine. IFN-γ and IL-17 levels, determined by ELISA in a supernatant of lung tissue homogenate were analyzed. The IFN-γ levels and IL-17A levels (E) in spleen at 12 hours post-challenge on day 7 and day 112 after the second vaccination were determined by ELISA. Data are presented as means ± SEM. Significant differences were calculated with One-way ANOVA followed by Tukey's multiple comparisons test. *p < 0.05, **p < 0.01, ***p < 0.001.
(TIF)

**S6 Fig. Amount of antigen-specific IgG in the recipient mice and actively immunized mice.** Antigen-specific IgG in serum of the recipient mice and actively immunized mice was detected using ELISA. Data are presented as means ± SEM. Significant differences were calculated with One- or Two-way ANOVA followed by Tukey's multiple comparisons test. ns, not significant, *p < 0.05, ***p < 0.001.
(TIF)

## Acknowledgments

We thank Dr. Z. Yin (College of Life Sciences, Jinan University, Guangzhou, China) for kindly providing the TCR δ$^{-/-}$ (JAX Stock number: 003288), *il-17*$^{-/-}$ (NCBI Stock number: 16171) and *ifn-γ*$^{-/-}$ (JAX Stock number: 002116) mice.

## Author Contributions

**Conceptualization:** Haochi Zhang, Shouxin Sheng, Xiao Wang.

**Data curation:** Haochi Zhang, Shouxin Sheng, Chunhe Li, Xuemei Bao, Lixia Zhao, Jian Chen, Pingyuan Guan, Xiaoyan Li, Na Pan, Yanchen Liang, Xueqi Wang, Jingmin Sun, Xiao Wang.

**Formal analysis:** Haochi Zhang, Shouxin Sheng, Chunhe Li, Xuemei Bao, Na Pan, Yanchen Liang, Xueqi Wang, Jingmin Sun.

**Funding acquisition:** Haochi Zhang, Xiao Wang.

**Investigation:** Haochi Zhang, Shouxin Sheng, Xiao Wang.

**Methodology:** Haochi Zhang, Shouxin Sheng, Chunhe Li, Xuemei Bao, Na Pan, Yanchen Liang, Xueqi Wang, Jingmin Sun.

**Project administration:** Haochi Zhang, Shouxin Sheng, Chunhe Li, Xuemei Bao, Yanchen Liang, Xueqi Wang, Jingmin Sun.

**Resources:** Haochi Zhang, Xiao Wang.

**Software:** Haochi Zhang, Shouxin Sheng, Chunhe Li, Xuemei Bao, Yanchen Liang.

**Supervision:** Haochi Zhang, Lixia Zhao, Jian Chen, Pingyuan Guan, Xiaoyan Li, Xiao Wang.

**Validation:** Lixia Zhao, Jian Chen, Pingyuan Guan, Xiaoyan Li, Na Pan, Xiao Wang.

**Visualization:** Haochi Zhang, Shouxin Sheng, Chunhe Li, Xuemei Bao, Na Pan, Yanchen Liang, Xueqi Wang, Jingmin Sun.

**Writing – original draft:** Haochi Zhang, Shouxin Sheng, Na Pan, Yanchen Liang.

**Writing – review & editing:** Haochi Zhang, Shouxin Sheng, Chunhe Li, Xuemei Bao, Lixia Zhao, Jian Chen, Pingyuan Guan, Xiaoyan Li, Na Pan, Yanchen Liang, Xueqi Wang, Jingmin Sun, Xiao Wang.

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
