## [Decision Letter · Decision Letter 0]

5 Apr 2024

Dear Dr. Wang,

Thank you very much for submitting your manuscript "Mucosal immunization with the lung Lactobacillus-derived amphiphilic exopolysaccharide adjuvanted recombinant vaccine improved protection against P. aeruginosa infection" for consideration at PLOS Pathogens.Your manuscript was reviewed by members of the editorial board and by several independent reviewers. In light of the reviews (below), we would like to invite the resubmission of a significantly revised version that takes into account the reviewers' comments.

The reviewers found your work potentially interesting, but they raised concerns about the advance your findings represent over earlier work (particularly your previous paper describing the same adjuvant) and the strength of the novel conclusions that can be drawn at this stage. All reviewers agreed that the potential novelty would be the evaluation of a vaccine composed of PcrV and Lactobacillus-derived exopolysaccharide as an adjuvant. However, referees had many concerns regarding the results as well as in vivo models and underlined some weaknesses in the data you provided

The first major issue raised by all reviewers is the challenge dose/administration reported in the manuscript. The results in Figure 1 and 4 should be more described and discussed.

The second major issue is the difference between the results in Figure 2 and 3. The authors should better explain and discuss these points.

Finally, the last major issue concerns the infection model. As pointed out by reviewer #3, use of PA01 rather than a T3SS-producing strain should be discussed since lower virulence might impact vaccine efficacy evidencing. In addition, the choice of germline Tcrd -/- mice should be justified.

We cannot make any decision about publication until we have seen the revised manuscript and your response to the reviewers' comments. Your revised manuscript is also likely to be sent to reviewers for further evaluation.

Sincerely,

Thomas Guillard, PharmD, PhD

Academic Editor

PLOS Pathogens

D. Scott Samuels

Section Editor

PLOS Pathogens

Michael Malim

Editor-in-Chief

PLOS Pathogens

orcid.org/0000-0002-7699-2064

From the reports, you will see that while they found your work of some potential interest, the referees raise concerns about the advance your findings represent over earlier work (particularly your previous paper describing the same adjuvant) and the strength of the novel conclusions that can be drawn at this stage. All reviewers agreed that the potential novelty would be the evaluation of a vaccine composed of PcrV and Lactobacillus derived exopolysaccharide as an adjuvant. However, referees had many concerns regarding the results as well as in vivo models and underlined some weaknesses in the data you provided

The first major issue raised by all reviewers is the challenge dose/administration reported in the manuscript. The results in Figure 1 and 4 should be more described and discussed.

The second major issue is the difference between the results in Figure 2 and 3. The authors should better explained and discussed these points.

Eventually, the last major issue concerns the infection model. As pointed out by reviewer #3, use of PA01 rather than a T3SS-producing strain should be discussed since lower virulence might impact vaccine efficacy evidencing. In addition, the choice of germline Tcrd -/- mice should be argued in support.

Reviewer's Responses to Questions

**Part I - Summary**

Reviewer #1: The manuscript of Kiao Wang et al reports on nasal immunisation against P aeruginosa using an encapsulating lactobacillus exo poly saccharide as adjuvant, that was compared to Cholera toxin adjuvant. Ab induction including pulmonary IgA were checked as well as protection efficacy in either direct of adoptive cell transfer challenge tests. Cellular response was assessed by cytokine dosages and flow cytometry analysis of lung and spleen cells populations. Immunisation and challenge of IFN, IL17, Gamma delta T cell deficient mice were also performed. Displaying an interesting effect of IL17 deficiency on P aeruginosa infection.

Although reporting an interesting work, several points of the manuscript need to be clarified or better presented.

Reviewer #2: The authors evaluated protection efficacy and mechanism of a vaccine composed of PcrV and Lactobacillus derived exopolysaccharide as an adjuvant (EPS301@rPcrV). The vaccine conferred protection against P. aeruginosa lung infection. The authors further demonstrated that the vaccine elicited antibody against PcrV. By adaptive transfer and using IL-17A knockout mice, they demonstrated that CD4 and γδ T cells mediated IL-17 production play important roles in the protection.

Reviewer #3: The paper by Zhang et al report on a Lactobacillus eps vaccine that elicits IgA as well as lung Th17 and gd 17+ cells. This vaccine provides protection against a PA01 challenge. The data are interesting but preliminary. Moreover, the cellular immunology is not state of the art.

**Part II – Major Issues: Key Experiments Required for Acceptance**

Reviewer #1: = Figure1 I (imaging) is not supporting comments done in the main text. Firstly, amounts (or signal settings) of fluorescent OVA appear to be different for OVA alone or encapsulated OVA. Secondly, I do not agree with the given interpretation of images (look at 12 hours): although the free OVA is only located in one lung (suggesting a direct bronchial and not nasal administration), adjuvanted OVA is mostly found in the gut or bladder, and also in the liver for a small part. And incidentally not in the spleen throughout the whole experiment. Suggesting rapid clearance and/or degradation of most of the Ag. No data using the true P aeruginosa recombinant Ag are provided. That figure should be either completed by another experiment, more extensively commented, or deleted.

= I'm unable to understand the difference between survival curves in fig 2 and fig 3. Please clarify.

Reviewer #2: 1. Fig. 1I: At 12 hours post the inoculation, the Cy7 signal in the lung is stronger in the mouse received Cy7-OVA than that received EPS301@Cy7-OVA. It seems Cy7-OVA retain longer in the lung, which should be discussed. Meanwhile, which organs are the Cy7-OVA distributed after 12 hours? It does not seem to be in the spleen.

2. Fig. 4: What was the challenge dose?

Fig. 4A, B: The IFN-γ+ CD4+ T cells in all the samples in the day 112 group were higher than the corresponding ones in the day 7 group, however, the IL-17A+ CD4+ T cells were lower. Why the IFN-γ response was stronger after longer period of time after vaccination? These results need to be confirmed and explained.

The same question for the Fig. 4C, D: the IFN-γ+ γδ+ T cells and IL-17A+ γδ+ T cells were higher in the day 112 groups than those in the day 7 groups.

The overall IFN-γ level was lower in the CTB+ rPcrV immunized mice in the day 112 groups than that in the day 7 group (Fig. 4E), and the IFN-γ level was similar in the EPS301@rPcrV immunized mice in the day 112 groups than that in the day 7 group. These results are contradictory to the higher numbers and percentages of IFN-γ+ cells in the day 112 groups (Fig. 4A, 4C).

3. Fig. 5A: When were the sera collected (how many days post the second immunization)?. Based on the method description, after incubation with PAO1, the extracellular bacteria were washed away and the macrophage cells were lysed, followed by determination of live intracellular bacterial numbers by plating. Thus the experiment was the opsonophagocytosis assay, not opsonophagocytosis killing activity as stated in the figure legend.

If this was the opsonophagocytosis killing assay, it is strange that the bacterial numbers remained similar after dilution of the sera from immunized mice when 10e4 CFU of PAO1 was used.

4. Fig. 5C. A previous report demonstrated that anti-PcrV IgG was able to confer protection in mice (Nat Med. 1999. 5(4):392-8. doi: 10.1038/7391). An explanation should be provided.

5. For the T cell transfer assays, why were the numbers of γδ T cells (10000) and CD4+ T cells (50000) different?

6. Transfer of CD4+ T cells resulted in approximately 10-fold less bacteria in the lungs than those received γδ T cells (Fig. 5H and 6F), however, the IL-17A levels was similar between the two groups (Fig. 5I and 6G). Does it mean that besides IL-17 the CD4+ T cells provided other mechanisms of protection?

Reviewer #3: 1. The IVIS imaging shows a lot of the vaccine is swallowed or administered down the esophagus. This is likely due to the large volume administered. Was the 40 uL delivered in a single nare or split across left and right nare? Volumes above 15 ul are more likely to aspirated into the lung/stomach. Also what anesthesia was used?

2. The authors use a PA01 challenge strain. It is my understanding this strain does not efficiently express the Type III secretion system and some labs have this used PA103 or PA14 to model ventilator associated pneumonia which is associated with Type III secretion.

3. It unclear if the authors are eliciting lung TRM cells. This would need to be defined by exclusion of anti-CD45 IV staining. Additionally TRM cells are CD69+, CXCR6+, S1PR1 lo, KLF2 lo. If these cells are elicited – do they contract over time?

4. Germline Tcrd-/- mice have known developmental differences in alpha-beta cells. The authors would need to show that Th17 cells and IgA are still present in these mice.

**Part III – Minor Issues: Editorial and Data Presentation Modifications**

Reviewer #1: = Size of nano-particles of EPS301 encapsulating protein is far bigger than that of EPS301 alone but data are from BSA and not from the true Ag used and a single ratio of EPS versus protein seems to have been used (lines574-579). It would be of interest to look at the change of particle size according to various ratios or to the constant size due to mechanical self assembly constraints.

=It would be mandatory to give details of Ag concentration/ mouse and Ag/adjuvant ratio as well ( and not those of BSA as the developement model.

= lung pathological findings of challenged mice are provided, but not from immunized mice before any challenge. that would also be of interest.

=Fig 3 F (12h challenge) deserves more comments as supporting efficacy of the proposed vaccine: if pictures are representative, Ag alone is poorly protective whereas EPS301@rPcrV is totally protective. Cholera toxin exhibits a global inflammation. As mentioned above, pathological samples without challenge would have allowed to decipher between something due to Cholera toxin itself or a different answer to P aeruginosa challenge after use of CTB.

Reviewer #2: Fig. 2D. The scale bar is missing.

Line 426 No need to use the full name “Pseudomonas aeruginosa” here.

Reviewer #3: n/a

PLOS authors have the option to publish the peer review history of their article (what does this mean?). If published, this will include your full peer review and any attached files.

Reviewer #1: No

Reviewer #2: No

Reviewer #3: No
---

## [Decision Letter · Decision Letter 1]

21 Jul 2024

Dear Dr. Wang,

Thank you very much for submitting your manuscript "Mucosal immunization with the lung Lactobacillus- derived amphiphilic exopolysaccharide adjuvanted recombinant vaccine improved protection against P. aeruginosa infection" for consideration at PLOS Pathogens. As with all papers reviewed by the journal, your manuscript was reviewed by members of the editorial board and by several independent reviewers. In light of the reviews (below this email), we would like to invite the resubmission of a significantly-revised version that takes into account the reviewers' comments.

I have recommended "Major revisions" while two referees recommended "Minor revisions" because
I am surprised by the variety of responses in your rebuttal. Some of them carefully addressed the reviewer's questions, while others had significant fundamental flaws. I agree with reviewer #1 that it is puzzling using cropped NIR in vivo imaging with a product move (OVA instead of the true Ag) from the nose to the lung at 12 hours, despite no signal in this area at T0 or T6h. 

We cannot make any decision about publication until we have seen the revised manuscript and your response to the reviewers' comments. Your revised manuscript is also likely to be sent to reviewers for further evaluation.

Sincerely,

Thomas Guillard, PharmD, PhD

Section Editor

PLOS Pathogens

D. Scott Samuels

Section Editor

PLOS Pathogens

Michael Malim

Editor-in-Chief

PLOS Pathogens

orcid.org/0000-0002-7699-2064

Reviewer's Responses to Questions

**Part I - Summary**

Reviewer #1: Several comments have been addressed, leading to changes improving the manuscript. But some points remain difficult to understand and others could be further improved.

Reviewer #2: The authors have adequately addressed critical comments.

Reviewer #3: The authors have addressed by concerns.

**Part II – Major Issues: Key Experiments Required for Acceptance**

Reviewer #1: I'm focusing on the answers given to reviewer one comments.

R 1 major comments.

= Comment 1 dealing with figure 1. The response refers to the text ligne 148-151 as further explanation given. "In addition, EPS301 encapsulation could effectively penetrate the nasal mucous layer and prolonged antigen retention " That only sentence can hardly make clear the new data from the Figure 1 E On that figure announced from new experiments, no signal is observed with OVA, whereas the signal from EPS301@ova in the lung was detected from T 12 hours and negative before. On the left part (images), whole mouse body imaging should be given, at least for the most significant time points, instead of cropped images. To be sure that signal is from the lungs and not liver or spleen and to illustrate the relative amount ending in the guts.

= concerning the difference between survival curves in fig 2 and 3. Explanations are given in the answer to the comment but not in the manuscript. Moving that point as a minor issue now. BTW, are the 2 missing curves on fig 2B totally negatives or not done ?

Reviewer #2: (No Response)

Reviewer #3: In the statistical analysis section, the authors should specify why they used ANOVA vs Kruskal-Wallis. Also which multiple comparisons test was used such as Dunn's, Hol-Sidak etc. If different tests were used with specific Figures- this information should be in the Figure Legend.

**Part III – Minor Issues: Editorial and Data Presentation Modifications**

Reviewer #1: = comment 1 concerning the size of the particles. Additional information have been provided and the choice of the 1mg/ml concentration for particle synthesis justified. If the I. N. administration was done at twice that concentration, the description of immunogen preparation should mention it and how the concentration was done.

= comment 2 concerning amount of Ag, adjuvant and ratio. The sentence added in the manuscript does not answer the comment: ""intranasal (i.n.) route (volume 40 μl, antigen, 2mg/kg, and EPS301, 2mg/kg)"" I can't believe that doses have been calculated from the weight of each mice. If kg is a mistyping for ml, please correct.

= comment 3 Histology of lungs before any challenge. New data provided demonstrate minimal inflammation in the EPS301@rPcrV group or no inflammation in the other. That should be kept in line with the inflammation obtained post challenge from CTB (see comment4 and answer) and deserves in the manuscript not only a comment but some discussion on the mechanism involved.

Yellow marks:

L 27-28 OK

L 43-49 OK

L138-141 OK minor fig 1B left TEM of EPS301 If available, a snapshot with a better focused beam should be better

L153-191 OK but some concerns on the figure E

L239-244 L249-265 OK

L268-279 OK

L289-295 OK

L315-326 OK

L348-354 OK but nature and rationale for FTY720 (fingolimod) use is not given in the manuscript. Amount used is also not given. All readers are not familiar with selective immunosuppression...

L442 OK A mention of the severity of ocular infection with P aeruginosa could be added

L509-515 OK

L531-536 OK but the mention of "several papers" lead to add at least the reference of a review

L539-542 OK "permanent"could be added before elevated levels lof IL17

L605 OK

L624-625 NO Cf above

L656-657 OK btw L658, expression of FCS as 5% final is more usual..

L674-675 OK

L693-895 OK the rationale for the use of anti Fc gamma receptors might be given

L699-703 OK

Reviewer #2: (No Response)

Reviewer #3: (No Response)

PLOS authors have the option to publish the peer review history of their article (what does this mean?). If published, this will include your full peer review and any attached files.

Reviewer #1: No

Reviewer #2: No

Reviewer #3: No
---

## [Decision Letter · Decision Letter 2]

16 Oct 2024

Dear Dr. Wang,

Thank you very much for submitting your manuscript "Mucosal immunization with the lung Lactobacillus- derived amphiphilic exopolysaccharide adjuvanted recombinant vaccine improved protection against P. aeruginosa infection" for consideration at PLOS Pathogens. The reviewer appreciated the attention to an important topic. Based on the reviews, we are likely to accept this manuscript for publication, providing that you modify the manuscript according to the review recommendations.

As you will see, the reviewer proposes to add a minor revision for a better comment regarding the findings of a secondary move to NIR signal to the lung. 

Sincerely,

Thomas Guillard, PharmD, PhD

Section Editor

PLOS Pathogens

D. Scott Samuels

Section Editor

PLOS Pathogens

Michael Malim

Editor-in-Chief

PLOS Pathogens

orcid.org/0000-0002-7699-2064

Reviewer Comments (if any, and for reference):

Reviewer's Responses to Questions

**Part I - Summary**

Reviewer #1: Authors performed new experiments to comply with referee's queries, displaying new interesting features. Details have been also corrected like quality of some images. The blocking point of the fig 1 is now solved.

The manuscript is now acceptable for publication.

**Part II – Major Issues: Key Experiments Required for Acceptance**

Reviewer #1: I would only ask for a minor modification. New in vivo imaging clearly show an unexpected kinetics following nose administration of the encapsulated immunogen. The same time there is some better nose retention than using Ag alone, a secondary move of the NIR signal to the lung after one day is observed. No comments on the potential mechanism of that phenomenon is done. I would propose two hypothesis to discuss: a) a drying of the product instillated in the nose with a subsequent inhalation of crusts when breathing. a) a nose obstruction due to a local reaction, leading to some snoozing with inhalation at the inspiratoty phase of coughing.

The difference in protection according to the time is now better depicted and correctly linked to the problem of reaching an enduring immune response, rightly considered to be a major axis for future works.

**Part III – Minor Issues: Editorial and Data Presentation Modifications**

Reviewer #1: Minors issues raised in the previous version of the manuscript have been correctly addressed.

PLOS authors have the option to publish the peer review history of their article (what does this mean?). If published, this will include your full peer review and any attached files.

Reviewer #1: No

Figure Files:

Data Requirements:

Reproducibility:

References:

---

## [Editor Report · Decision Letter 3]

25 Oct 2024

Dear Dr. Wang,

We are pleased to inform you that your manuscript ' Mucosal immunization with the lung Lactobacillus- derived amphiphilic exopolysaccharide adjuvanted recombinant vaccine improved protection against P. aeruginosa infection' has been provisionally accepted for publication in PLOS Pathogens.

Best regards,

Thomas Guillard, PharmD, PhD

Section Editor

PLOS Pathogens

D. Scott Samuels

Section Editor

PLOS Pathogens

Michael Malim

Editor-in-Chief

PLOS Pathogens

orcid.org/0000-0002-7699-2064
---

## [Editor Report · Acceptance letter]

11 Nov 2024

Dear Dr. Wang,

We are delighted to inform you that your manuscript, " Mucosal immunization with the lung Lactobacillus- derived amphiphilic exopolysaccharide adjuvanted recombinant vaccine improved protection against P. aeruginosa infection," has been formally accepted for publication in PLOS Pathogens.

Best regards,

Michael Malim

Editor-in-Chief

PLOS Pathogens

orcid.org/0000-0002-7699-2064